# Just Noticeable Difference Modeling for Deep Visual Features

Rui Zhao[1]   Wenrui Li[2]   Lin Zhu[3]   Yajing Zheng[4]   Weisi Lin[1]

## Abstract

Deep visual features are increasingly used as the interface in vision systems, motivating the need to describe feature characteristics and control feature quality for machine perception. *Just noticeable difference* (JND) characterizes the maximum imperceptible distortion for images under human or machine vision. Extending it to deep visual features naturally meets the above demand by providing a task-aligned tolerance boundary in feature space, offering a practical reference for *controlling feature quality* under constrained resources. We propose *FeatJND*, a task-aligned JND formulation that predicts the maximum tolerable per-feature perturbation map while preserving downstream task performance. We propose a FeatJND estimator at standardized split points and validate it across image classification, detection, and instance segmentation. Under matched distortion strength, FeatJND-based distortions consistently preserve higher task performance than unstructured Gaussian perturbations, and attribution visualizations suggest FeatJND can suppress non-critical feature regions. As an application, we further apply FeatJND to token-wise dynamic quantization and show that FeatJND-guided step-size allocation yields clear gains over random step-size permutation and global uniform step size under the same noise budget. The source code is available at https://github.com/ruizhao26/FeatJND.

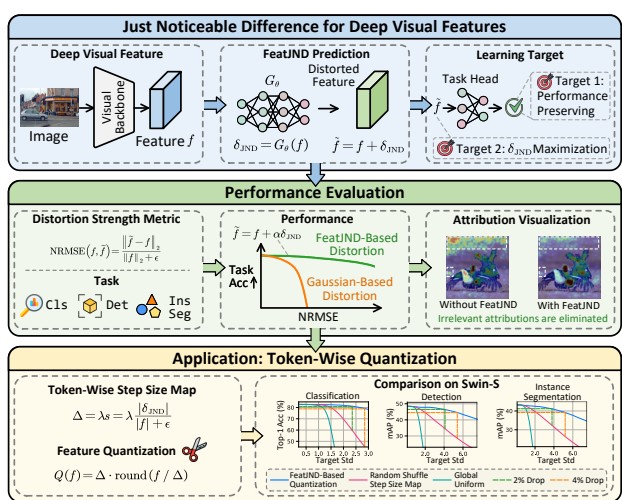

*Figure 1.* Illustration of the just noticeable difference for deep visual features (FeatJND). In this paper, we first construct the concept of FeatJND, and then evaluate it on different tasks, and verify its effectiveness on a token-wise quantization application, which means spatially-varying step sizes shared across channels.

## 1. Introduction

As computer vision models and systems rapidly evolve, an increasing number of applications no longer treat raw images as the only exchangeable entity. Instead, deep visual features are becoming a common interface both within and across systems (Gao et al., 2025b). In edge-cloud deployments, features are extracted on-device and transmitted for downstream inference (Wang et al., 2024b); in foundation-model and multi-task pipelines, modules are composed and reused through deep features (Davari & Belilovsky, 2024); and in distributed training and inference, intermediate states often become a dominant bottleneck (Huang et al., 2019). As a result, features are emerging as the "information currency" of modern vision systems. They carry the semantic and structural content that governs task performance, while simultaneously driving bandwidth, latency, memory, and compute costs. This motivates the need for a task-aligned tolerance threshold that characterizes the maximum feature distortion permissible without inducing a perceptible degradation in downstream performance. *This can also guide us to control feature quality under constrained resources.*

In practical feature-based vision systems, we often need to compress (Gao et al., 2025a), quantize (Huang et al.,

[1]College of Computing and Data Science, Nanyang Technological University, Singapore [2]Department of Computer Science and Technology, Harbin Institute of Technology, Harbin, China [3]School of Artificial Intelligence, Beijing Normal University, Beijing, China [4]School of Computer Science, Peking University, Beijing, China. Correspondence to: Weisi Lin <wslin@ntu.edu.sg>.

*Proceedings of the 43$^{rd}$ International Conference on Machine Learning*, Seoul, South Korea. PMLR 306, 2026. Copyright 2026 by the author(s).

2024), or downsample intermediate features to meet system constraints such as bandwidth and latency. A simple approach is to measure feature distortion using simple metrics, e.g., mean squared error (MSE). However, such metrics are often insufficient to reflect the task-level impact. Distortions of the same magnitude can lead to drastically different drops in downstream accuracy depending on where they occur in the representation, while in other cases, large numeric discrepancies may leave the final prediction almost unchanged. In other words, the relationship between feature-space distortion and task performance is not stable, making an "acceptable distortion boundary" difficult to define. As a result, system-level resource allocation and quality control are often driven by heuristic tuning, which can limit the interpretability and generalization across tasks and models.

Essentially, we need a threshold-based characterization of feature quality for machine vision. In the image domain, perceptual thresholding under the human vision system (HVS) provides a canonical paradigm: **Just Noticeable Difference (JND)** (Lin & Ghinea, 2021) describes the boundary at which distortions become just perceptible to humans. Building on this, several works have developed methods to predict JND maps (Jiang et al., 2022; Wang et al., 2025) and to leverage them in vision tasks (Wang et al., 2016; Zhang et al., 2021b), where they allocate more bits or computation to perceptually sensitive regions. The success of JND highlights the value of threshold-based formulations: they offer an operational notion of a quality boundary that can directly guide system-level resource allocation.

Traditional JND is defined for image distortions under human perception, while in machine vision, the relevant perception is reflected by downstream task performance. Some recent studies explore JND for machine vision (Jin et al., 2021; Zhang et al., 2021a), typically by defining imperceptibility through negligible changes in task metrics. However, they remain in the image domain. As deep features become the objects that are directly transmitted, reused, and compressed, a key question arises: *Does a JND-like tolerance boundary also exist in deep feature space?*

In this paper, we explore Feature-JND (FeatJND) to model the feature quality boundary for machine perception. Given a feature and its downstream task, we predict a FeatJND map as a perturbation of the feature, where we aim to keep the task performance drop within a small tolerance. To probe the machine-perception boundary, we also require the predicted FeatJND perturbation to be as large as possible while remaining imperceptible to the downstream task. Our contributions can be summarized as follows.

**(1)** We propose FeatJND, a just noticeable difference formulation for deep visual features that predicts the maximum tolerable perturbation while preserving downstream outputs.

**(2)** We develop a simple but effective FeatJND estimator and validate it on image classification, detection, and instance segmentation with diverse backbones.

**(3)** We show the effectiveness of FeatJND via matched-distortion-strength evaluations, attribution visualizations, and an application in token-wise feature quantization.

## 2. Related Work

**Just Noticeable Difference (JND).** JND originated from the Human Visual System (HVS) (Jayant et al., 2002), aiming to characterize the boundary at which distortions become just perceptible. In the image domain, several works have explored obtaining fine JND maps, ranging from modeling of visual mechanisms (Wu et al., 2017; Chen & Wu, 2019) to data-driven approaches that incorporate semantic cues (Wang et al., 2024a), frequency information (Wang et al., 2025), etc. The JND maps have been adopted in applications such as image compression (Wang et al., 2016; Zhang et al., 2021b; Li et al., 2025) and image quality assessment (Seo et al., 2020) to distinguish and handle areas with different sensitivities. More recently, a few studies have explored JND from a machine-perception perspective (Jin et al., 2021; Zhang et al., 2021a), formulating the boundary as the maximum perturbation under a performance constraint. However, these efforts primarily focus on the image domain, and feature JND remains relatively underexplored.

**Feature Quality Modeling.** Feature-JND can be regarded as a way of characterizing feature quality. In recent machine-oriented image/video compression and collaborative inference research (Duan et al., 2020), several works have attempted to measure feature quality. Some works simply use the MSE between original and compressed features as the quality metric for system optimization (Choi & Bajić, 2018; Kim et al., 2023), assuming that smaller numeric deviation implies better representations. Some other works tie feature quality to downstream task performance, using the drop in task accuracy after compression to quantify feature quality (Alvar & Bajić, 2021; Feng et al., 2022). Both types of formulations typically depend on a specific network architecture and task setup, making them difficult to transfer and reuse across tasks and scenarios.

**Adversarial Attack.** Adversarial attacks study how to optimize a targeted perturbation to maximize the loss or induce erroneous predictions under given constraints, which can be categorized as white-box and black-box. In the white-box setting (Madry et al., 2018; Moosavi-Dezfooli et al., 2017; Athalye et al., 2018), the attacker has access to the model parameters and gradients to construct strong perturbations. In the black-box setting (Ilyas et al., 2018; Guo et al., 2019), the attacker cannot directly access parameters or gradients and needs to estimate attack directions by

querying the model outputs. ***Different from adversarial attacks***, FeatJND focuses on *redundancy characterization*. Instead of finding the *worst-case* perturbation to cross decision boundaries using model internals, we estimate the maximum *safe volume* for quality control. Crucially, unlike attacks that require backpropagation or model access, during inference, FeatJND serves as a *gradient-free* estimator that predicts tolerance maps solely from the feature itself.

## 3. Methodology

### 3.1. Description

Just noticeable difference (JND) characterizes the smallest change in a signal that becomes perceptible to human observers. It can equivalently be viewed as the maximum distortion that remains imperceptible under a given viewing condition. In classical psychophysics, JND is often approximated by generalized Weber's law (Lin & Ghinea, 2021), i.e., $\mathrm{JND}(X) = kX + \Delta X_\mathrm{s}$, where $X$ is the signal, $k$ is the Weber fraction depending on the stimulation type, and $\Delta X_\mathrm{s}$ is a spontaneous term for low-stimulus regimes. Since distortions below the JND threshold typically do not cause noticeable perceptual degradation, JND provides a principled basis for saving bandwidth, storage, or computation while preserving perceived quality.

Motivated by this paradigm, we extend the notion of "imperceptibility" from human visual systems to machine perception based on deep visual features. In our setting, the perception is defined by the downstream task performance based on the features. We therefore define *Feature JND (FeatJND)* in feature space as follows. Given an intermediate feature tensor $f$ at a system interface, i.e., a split point, we learn a FeatJND map $\delta$ that has the same shape as $f$ to make a distortion to $f$, where $\delta$ is the maximum distortion that can be tolerated without compromising the task performance. The JND map can be used for subsequent resource allocation, such as feature quantization and dropping.

We define "machine imperceptibility" as requiring task outputs to remain within a prescribed tolerance under feature perturbation $\delta$. In the following, we detail the FeatJND formulation, estimator architectures, and loss functions for the three evaluated tasks.

### 3.2. Formulation

Denote $f \in \mathbb{R}^{C \times H \times W}$ as a target intermediate feature tensor at a split point, where $C, H, W$ indicate the shape of the feature. Let $h(\cdot)$ be the downstream task head. We seek the *maximum* feature distortion $\delta$ that remains *imperceptible* to the machine, where imperceptibility is quantified by task performance. We denote $P_\mathrm{t}(h(f))$ as the task performance based on feature $f$. We model the estimation of FeatJND as

an optimization problem based on a chance constraint:

$$\max_{\delta} \ \mathcal{M}(\delta)$$
$$\text{s.t.} \ \Pr_{\delta \sim \mathcal{Z}} \Big( P_\mathrm{t}\big(h(f)\big) - P_\mathrm{t}\big(h(f + \delta)\big) \leq \varepsilon \Big) \geq 1 - \rho, \quad (1)$$

where $\mathcal{M}(\delta)$ is a characterization of the amplitude of $\delta$, such as its $\ell_2$ norm. $\Pr$ means probability. $\delta \sim \mathcal{Z}$ indicates that $\delta$ is drawn from distribution $\mathcal{Z}$. $\varepsilon$ is the maximum tolerable performance drop. $\rho$ is a factor to allow rare violations. In Eq. (1), the meaning of the probability in the constraint term is the probability of achieving imperceptibility.

The chance constraint in Eq. (1) is difficult to optimize directly. We introduce the violation:

$$v(\delta) = P_\mathrm{t}\big(h(f)\big) - P_\mathrm{t}\big(h(f + \delta)\big) - \varepsilon, \quad (2)$$

and replace the probabilistic constraint by a hinge-based soft constraint:

$$\mathbb{E}_{\delta \sim \mathcal{Z}} \big[ [v(\delta)]_+ \big] \leq \tau, \quad (3)$$

where $[x]_+ = \max(x, 0)$ and $\tau$ controls the allowed expected violation. This yields:

$$\max_{\delta} \ \mathcal{M}(\delta) \quad \text{s.t.} \ \mathbb{E}_{\delta \sim \mathcal{Z}} \big[ [v(\delta)]_+ \big] \leq \tau. \quad (4)$$

We apply a Lagrangian relaxation to Eq. (4):

$$\max_{\delta} \ \min_{\lambda \geq 0} \ \mathcal{M}(\delta) - \lambda \Big( \mathbb{E}_{\delta \sim \mathcal{Z}} \big[ [v(\delta)]_+ \big] - \tau \Big), \quad (5)$$

which leads to the standard saddle-point training form by exchanging $\min$ and $\max$. In practice, we obtain the FeatJND map through a learnable $G_\theta$, leading $\delta = G_\theta(f)$, where $\theta$ denotes the parameters of the FeatJND estimator. Then, the optimization problem in Eq. (5) can be written as:

$$\min_{\lambda \geq 0} \ \max_{\theta} \ \mathcal{M}\big(G_\theta(f)\big)$$
$$- \lambda \mathbb{E}_{G_\theta(f)} \Big( \big[ P_\mathrm{t}(h(f)) - P_\mathrm{t}(h(f + G_\theta(f))) - \varepsilon \big]_+ \Big), \quad (6)$$

where $G_\theta(f) \sim \mathcal{Z}$ is omitted for brevity hereafter. Moreover, $P_\mathrm{t}(\cdot)$ is often non-smooth and non-differentiable, hence we replace $P_\mathrm{t}(\cdot)$ with a differentiable task discrepancy function $D_\mathrm{t}(f_1, f_2)$ computed on comparison between task-relevant outputs based on feature $f_1$ and $f_2$. Finally, although Eq. (6) suggests a primal-dual optimization over $(\theta, \lambda)$, in deep networks, the objective is non-convex, and strong duality is not guaranteed. We therefore adopt the common practice of fixing $\lambda$ as a hyperparameter, which yields a two-term objective. Based on the above analysis, the optimization object of the FeatJND estimation is:

$$\max_{\theta} \ \mathcal{M}\big(G_\theta(f)\big) - \lambda \mathbb{E}_{G_\theta(f)} \Big( D_\mathrm{t}\big(f + G_\theta(f), \ f\big) \Big). \quad (7)$$

Based on Eq. (7), FeatJND estimation can be viewed as finding a trade-off point between maximizing disturbance and minimizing task output variation. Therefore, there are 3 important sub-issues for FeatJND: (i) the selection of the interface feature $f$ at the split point; (ii) the architecture of the estimator $G_\theta$; and (iii) the design of a differentiable task discrepancy function $D_\mathrm{t}$. We will introduce our design for (i) and (ii) in Sec. 3.3 and (iii) in Sec. 3.4.

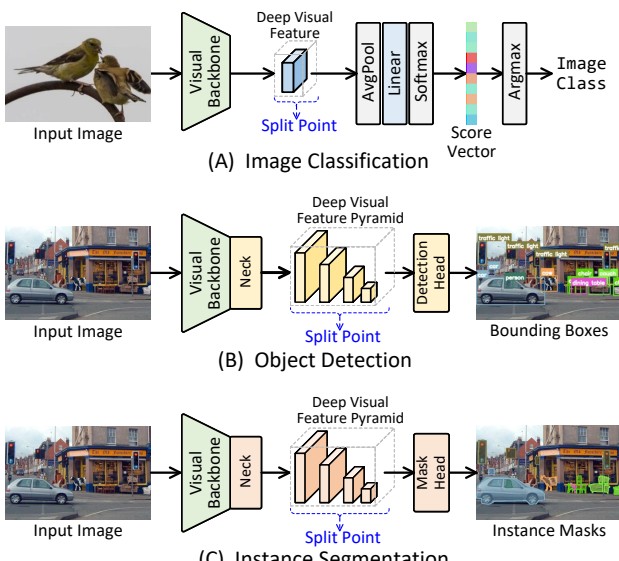

Figure 2. The split point selection of the downstream-task network.

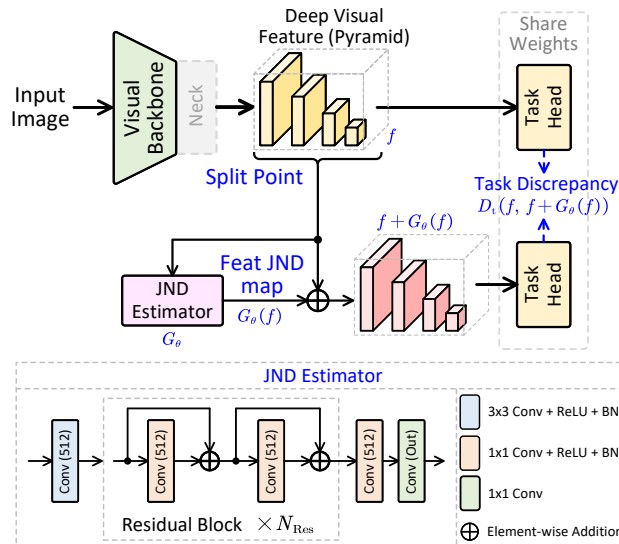

Figure 3. Training scheme and architecture of the JND estimator. BN means batch normalization.

## 3.3. Network Architecture

This subsection describes two components of our architecture: (i) the selection of the split point for the downstream task networks, which determines where we extract the target feature $f$ for FeatJND analysis; and (ii) the FeatJND estimator $G_\theta$, which predicts a JND map from $f$ to construct controlled feature distortions. We evaluate our framework on three representative vision tasks, namely image classification, object detection, and instance segmentation.

**Split point selection.** For image classification, we place the split point at the output of the visual backbone. For object detection and instance segmentation, we place the split point after the neck (e.g., FPN (Lin et al., 2017)), since the neck is a standardized feature aggregator that enriches backbone representations with multi-scale context and produces the feature interface consumed by task heads. Our selection is motivated by two considerations. *First*, backbone features form the most general-purpose and transferable deep visual representations, carrying semantic and structural information that is shared across tasks. *Second*, fixing the split point at the backbone or neck output provides a consistent interface across tasks and across different backbones.

Concretely, as shown in Fig. 2, for image classification, $f$ is taken from the backbone's final feature map and then passed to the classification head to obtain class logits. For object detection and instance segmentation, we similarly extract $f$ from the neck output and use it as the input to task heads, which produce class, box, and mask predictions.

**FeatJND estimator.** As an initial study, we adopt a simple yet effective convolutional estimator for $G_\theta$, whose architecture is shown at the bottom of Fig. 3. We aim to validate the feasibility of FeatJND. Thus, we do not explore more com-

plex network structures for FeatJND. Given $f$, $G_\theta$ predicts a same-shaped FeatJND map for controlled feature-space distortions. Specifically, $G_\theta$ consists of a $3 \times 3$ convolutional layer followed by a stack of residual blocks built with $1 \times 1$ convolutions, each with activation and normalization. We adopt $1 \times 1$ kernels since visual backbones typically downsample the input multiple times, yielding low-resolution features that already aggregate spatial information over local neighborhoods. For object detection and instance segmentation with multi-scale feature streams, we apply a shared $G_\theta$ to each feature level to predict the corresponding FeatJND maps without introducing additional architectural complexity.

Overall, this architectural design follows the objective in Sec. 3.2. The split point determines which intermediate representation we characterize to explore the tolerance boundary in feature space. Meanwhile, the estimator $G_\theta$ determines how this boundary is parameterized and estimated from $f$, yielding a FeatJND map for feature distortions. In the next subsection, we specify the task discrepancy function $D_\mathrm{t}$ for classification, detection, and instance segmentation, and derive the corresponding training losses.

## 3.4. Loss Function

Following the formulation in Eq. (7), our learning objective balances two competing goals: (i) enlarging the predicted distortion magnitude, characterized by $\mathcal{M}(G_\theta(f))$, and (ii) keeping the downstream-task output change small after applying the distortion, characterized by a task discrepancy term $D_\mathrm{t}(f, f + G_\theta(f))$. We use $\widetilde{f}_\theta = f + G_\theta(f)$ as an abbreviation for brevity hereafter. For training the JND estimator, we approximate the expectation by mini-batch sampling dur-

ing training. To comply with the standard "minimize-a-loss" paradigm, we minimize the negative of Eq. (7), yielding a two-term loss:

$$\mathcal{L} = \underbrace{\lambda_\mathrm{t}\, D_\mathrm{t}\big(\widetilde{f}_\theta, f\big)}_{\text{task consistency term}} - \underbrace{\mathcal{M}\big(G_\theta(f)\big)}_{\text{magnitude max term}}, \quad (8)$$

where $\lambda_\mathrm{t}$ controls the trade-off between preserving task outputs and increasing the distortion magnitude. Intuitively, the first term enforces that the downstream outputs remain consistent before and after applying the feature distortion, while the second term encourages larger distortions to better probe the tolerable boundary in the feature space.

A key challenge is the design of $D_\mathrm{t}$ for each task. It should be differentiable and reflect the output variations that matter for a given task. Since output structures differ significantly across tasks, we design task-specific definitions of $D_\mathrm{cls}, D_\mathrm{det}, D_\mathrm{ins}$ for image classification, object detection, and instance segmentation, respectively.

**Task discrepancy term for image classification.** For classification, we aim to keep the classifier prediction unchanged after applying the feature distortion. Let the classification head be $h_\mathrm{cls}$. The output logits vector for $f$ is $\mathbf{y} = h_\mathrm{cls}(f) \in \mathbb{R}^K$, where $K$ is the number of classes. The output logits vector for the distorted feature is $\widetilde{\mathbf{y}} = h_\mathrm{cls}(\widetilde{f}_\theta)$.

To measure the task discrepancy, we enforce consistency between the soft predictive distributions before and after distortion. Specifically, with a temperature $T$, we define

$$\mathbf{p} = \mathrm{softmax}(\mathbf{y}/T), \qquad \mathbf{q} = \mathrm{softmax}(\widetilde{\mathbf{y}}/T). \quad (9)$$

We then define the classification discrepancy term as the temperature-scaled KL divergence,

$$D_\mathrm{cls}(\widetilde{f}_\theta, f) = T^2\, \mathrm{KL}(\mathbf{p} \,\|\, \mathbf{q}) = T^2 \sum_{k=1}^{K} p_k \log \frac{p_k}{q_k}, \quad (10)$$

where we follow standard practice (Hinton, 2015) and multiply the KL terms by $T^2$ to preserve gradient magnitudes under temperature scaling. This term encourages the distorted feature to preserve the classifier's output distribution, where the temperature scaling provides smoother targets and stabilizes optimization.

**Task discrepancy term for object detection and instance segmentation.** For two-stage models such as Mask R-CNN (He et al., 2017), object detection and instance segmentation are produced in a single forward pass with shared components. This enables a decomposition of the task discrepancy term into a shared detection term and an additional mask term for instance segmentation. We generate proposals and regions of interest (ROIs) from the clean features and reuse them to evaluate both clean and distorted features, so the discrepancy is computed on aligned ROIs and is not confounded by ROI mismatching.

We design the detection discrepancy term to capture the stability of the two-stage detection pipeline, including the region proposal network (RPN) and the bounding-box head:

$$D_\mathrm{det}(\widetilde{f}_\theta, f) = D_\mathrm{cls}^\mathrm{rpn} + D_\mathrm{reg}^\mathrm{rpn} + D_\mathrm{cls}^\mathrm{roi} + D_\mathrm{reg}^\mathrm{roi}. \quad (11)$$

In particular, we measure classification-style consistency $\{D_\mathrm{cls}^\mathrm{rpn}, D_\mathrm{cls}^\mathrm{roi}\}$ via temperature-scaled KL divergence and regression-style consistency $\{D_\mathrm{reg}^\mathrm{rpn}, D_\mathrm{reg}^\mathrm{roi}\}$ via smooth $\ell_1$ norm (Huber, 1992) $\ell_{\mathrm{sm},1}$. Let $o$ be RPN objectness logits from $f$, $\mathbf{s}$ be ROI classification logits from $f$, and let $\mathbf{b}^\mathrm{rpn}$, $\mathbf{b}^\mathrm{roi}$ be the corresponding regression outputs from $f$. Analogously, $\widetilde{o}$, $\widetilde{\mathbf{s}}$, $\widetilde{\mathbf{b}}^\mathrm{rpn}$, and $\widetilde{\mathbf{b}}^\mathrm{roi}$ denote the respective outputs computed from the distorted feature $\widetilde{f}_\theta$. The discrepancy terms for detection can be formulated as:

$$D_\mathrm{cls}^\mathrm{rpn} = T^2\, \mathrm{KL}\Big(\mathrm{softmax}(o/T) \,\big\|\, \mathrm{softmax}(\widetilde{o}/T)\Big), \quad (12)$$

$$D_\mathrm{cls}^\mathrm{roi} = T^2\, \mathrm{KL}\Big(\mathrm{softmax}(\mathbf{s}/T) \,\big\|\, \mathrm{softmax}(\widetilde{\mathbf{s}}/T)\Big), \quad (13)$$

$$D_\mathrm{reg}^\mathrm{rpn} = \ell_{\mathrm{sm},1}\Big(\mathbf{b}^\mathrm{rpn}, \widetilde{\mathbf{b}}^\mathrm{rpn}\Big), \quad (14)$$

$$D_\mathrm{reg}^\mathrm{roi} = \ell_{\mathrm{sm},1}\Big(\mathbf{b}^\mathrm{roi}, \widetilde{\mathbf{b}}^\mathrm{roi}\Big), \quad (15)$$

where $T$ is a temperature parameter used for logit scaling in the KL divergence, producing softer probability distributions and more stable gradients.

Instance segmentation further requires the predicted masks to remain stable. We therefore add a mask-head consistency term, which is computed on the same ROIs. It can be formulated as:

$$D_\mathrm{ins}\Big(\widetilde{f}_\theta, f\Big) = D_\mathrm{det}\Big(\widetilde{f}_\theta, f\Big) + D_\mathrm{mask}\Big(\widetilde{f}_\theta, f\Big), \quad (16)$$

$$\text{where} \quad D_\mathrm{mask} = \mathrm{MSE}\big(\mathbf{m}, \widetilde{\mathbf{m}}\big), \quad (17)$$

where $\mathbf{m}$ and $\widetilde{\mathbf{m}}$ denote the mask logits produced by the mask head from $f$ and $\widetilde{f}$, respectively. Following classic works (Girshick, 2015; He et al., 2017), we avoid additional per-term coefficients inside $D_\mathrm{det}$ and $D_\mathrm{ins}$ for brevity.

## 4. Experiments

### 4.1. Experimental Settings

We evaluate FeatJND on three representative vision tasks, namely image classification, object detection, and instance segmentation, under feature-space distortions injected at split points in Sec. 3.3. For classification, we extract the target feature $f$ at the backbone output. For detection and instance segmentation, $f$ is taken from the neck output and fed into the corresponding heads. Table 1 summarizes the evaluated backbones, necks, and feature dimensions.

**Image classification.** We adopt both CNN- and Transformer-based classifiers, including ResNet-18/34/50 (He et al., 2016) and Swin Transformer (Swin-T/S/B) (Liu et al., 2021). All classification models are trained and evaluated on ImageNet-1K (Deng et al.,

*Table 1.* Models used in the experiments.

| Task | Index | Backbone | Neck | Feat. Ch. | Params (M) | Head |
|---|---|---|---|---|---|---|
| Classification | (A) | ResNet-18 | – | 512 | 11.7 | AvgPool + Linear |
| | (B) | ResNet-34 | – | 512 | 21.8 | |
| | (C) | ResNet-50 | – | 2048 | 25.6 | |
| | (D) | Swin-T | – | 768 | 28.3 | |
| | (E) | Swin-S | – | 768 | 49.6 | |
| | (F) | Swin-B | – | 1024 | 87.8 | |
| Det & Ins Seg | (G) | ResNet-50 | FPN | $\begin{bmatrix} 256 \\ 256 \\ 256 \\ 256 \end{bmatrix}$ | 44.4 | Mask R-CNN |
| | (H) | Swin-T | | | 47.8 | |
| | (I) | Swin-S | | | 69.1 | |
| | (J) | PVTv2-b1 | | | 33.7 | |
| | (K) | PVTv2-b2 | | | 45.0 | |
| | (L) | PVTv2-b3 | | | 64.9 | |

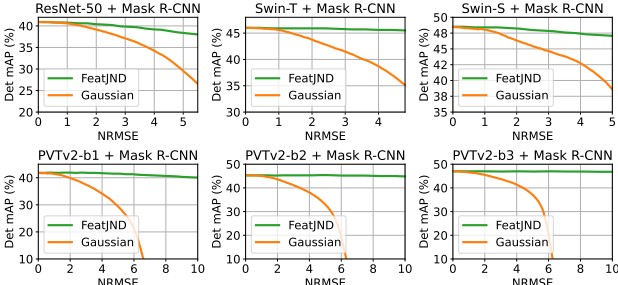

*Figure 5.* Task performance comparison for object detection between FeatJND-based and Gaussian noise distortions.

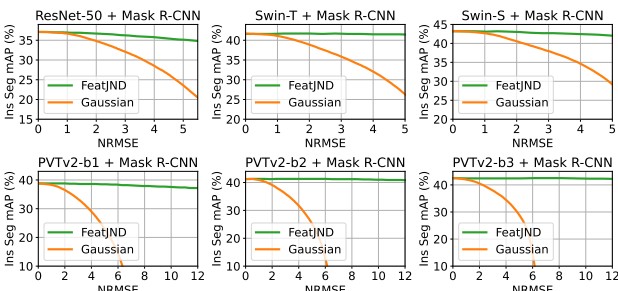

*Figure 6.* Task performance comparison for instance segmentation between FeatJND-based and Gaussian noise distortions.

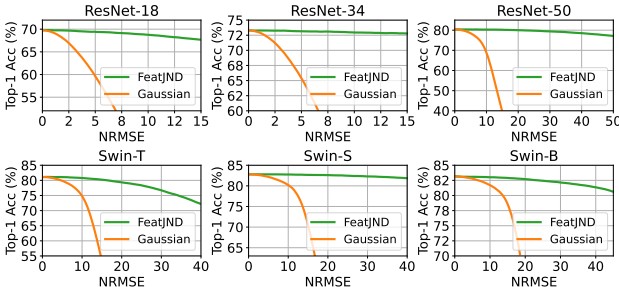

*Figure 4.* Task performance comparison for image classification between FeatJND-based and Gaussian noise distortions.

2009), which contains 1,281,167 training images and 50,000 validation images over 1,000 categories.

**Object detection and instance segmentation.** We instantiate a standard two-stage framework with a Mask R-CNN (He et al., 2017) head and an FPN (Lin et al., 2017) neck, and evaluate multiple backbones including ResNet-50, Swin-T/S, and different variants of PVTv2 (Wang et al., 2021; 2022). We conduct training on COCO2017 using the official `train2017` and `val2017` splits, which consists of 118,287 training images and 5,000 validation images over 80 object categories.

**Feature distortion metric.** To quantify the distortion introduced in the feature space, we measure the normalized root mean square error (NRMSE) between the clean feature $f$ and its distorted version $\widetilde{f}$. Since different backbones and tasks may produce features with different value scales, we normalize the distortion to ensure comparability across settings. Formally, we define:

$$\text{NRMSE}(f, \widetilde{f}) = \frac{\left\| \widetilde{f} - f \right\|_2}{\|f\|_2 + \epsilon}, \tag{18}$$

where $\epsilon$ is a small constant for numerical stability.

### 4.2. Implementation Details

For all tasks, we keep the downstream task network fixed with official pre-trained weights and only optimize the FeatJND estimator $G_\theta$. We use $\ell_2$ norm in Eq. (8) as the $\mathcal{M}(\cdot)$ function to describe the distortion amplitude.

**For image classification,** we train $G_\theta$ with a learning rate of `1e-4` for 30 epochs using a batch size of 128. We set $\lambda_t = 50$ to balance different loss terms to a comparable scale. We follow the standard ImageNet recipe with random resized cropping and horizontal flipping during training for augmentation. **For detection and instance segmentation,** we train $G_\theta$ with a learning rate of `2e-5` for 10 epochs using a batch size of 2. We set $\lambda_t = 200$ to balance different loss terms to a comparable scale.

**For all experiments,** we use the Adam optimizer to train $G_\theta$. The temperature in the task discrepancy loss is fixed to $T = 4$. We apply gradient clipping with a maximum $\ell_2$ norm of 1, and clamp the predicted distortion to $[-10, 10]$ for numerical stability.

### 4.3. Task Performance Measure for FeatJND

To validate that the FeatJND map $\delta_{\text{JND}} = G_\theta(f)$ meaningfully characterizes the *task-aligned* tolerance of different feature components, we compare FeatJND-based distortions with a zero-mean Gaussian noise baseline under NRMSE. Our key hypothesis is that $\delta_{\text{JND}}$ indicates where the feature can tolerate larger perturbations without causing noticeable degradation in downstream performance. Therefore, when the overall noise strength is controlled to be the same, the feature distorted according to $\delta_{\text{JND}}$ should yield higher task performance than an unstructured random perturbation.

**For FeatJND-based distortion,** given a clean feature $f$, we construct distorted features as $\widetilde{f} = f + \alpha\, \delta_{\text{JND}},\ \alpha > 0,$

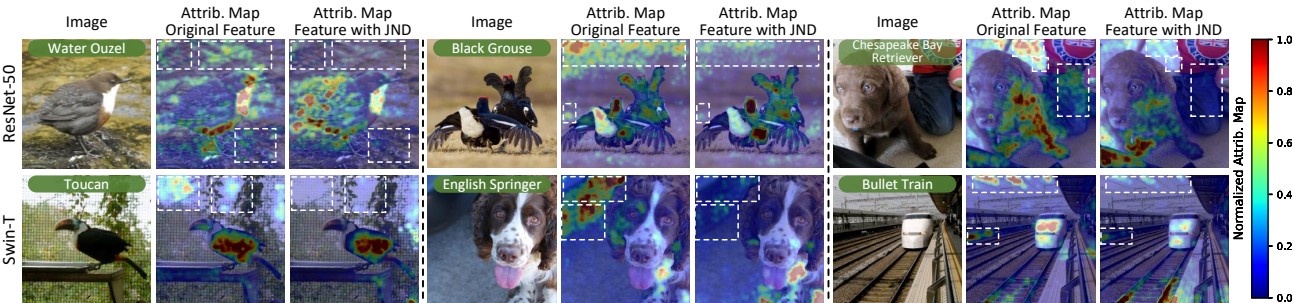

*Figure 7.* Comparison of attribution maps before and after applying FeatJND-based distortion on classification for ImageNet, which illustrates that FeatJND can reduce the influence of irrelevant areas while preserving important task-relevant regions in feature maps.

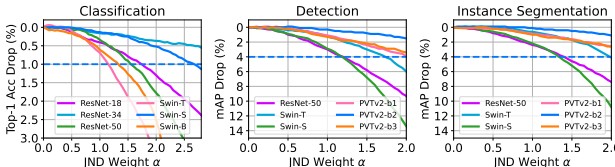

*Figure 8.* Performance drop of different models across three tasks as a function of the FeatJND distortion strength.

where $\alpha$ controls the distortion strength. **For Gaussian baseline,** we perturb the feature with i.i.d. Gaussian noise as $\tilde{f} = f + \epsilon, \ \epsilon \sim \mathcal{N}(0, \sigma^2 I)$, where we vary $\sigma$ to obtain different distortion levels. For the Gaussian baseline, *we use 5 random seeds*. For both distortion types, we measure the distortion strength using NRMSE (Eq. (18)) and compare the downstream task performance at the same NRMSE level. For object detection and instance segmentation with pyramid features, we apply distortion to each feature level and use the average NRMSE across all levels to measure distortion strength. Specifically, we report Top-1 accuracy for image classification and $\text{mAP}_{50:95}$ for object detection and instance segmentation, where $\text{mAP}_{50:95}$ denotes the mean average precision averaged over IoU thresholds from 0.50 to 0.95 with a step size of 0.05.

Fig. 4, Fig. 5, and Fig. 6 show task performance comparisons between FeatJND-based distortions and Gaussian noise. The values of Gaussian noise in the figures are the mean values calculated over 5 random seeds. Since the performance differences across different seeds are small, no shadow areas are plotted around the curves for the variance. Across all tasks and backbones, FeatJND-based distortions achieve higher performance than Gaussian noise at matched NRMSE, indicating that the predicted $\delta_{\text{JND}}$ captures the heterogeneous tolerance of different feature components and allocates perturbations to less task-sensitive parts. Experiments with more metrics are included in the Appendix.

Fig. 8 shows the performance drop across the three tasks as the JND weight $\alpha$ increases. It is shown that when $\alpha \leq 1$, the performance remains stable for most models, while noticeable performance degradation occurs when $\alpha > 1$. This suggests that the predicted $\delta_{\text{JND}}$ provides a calibrated

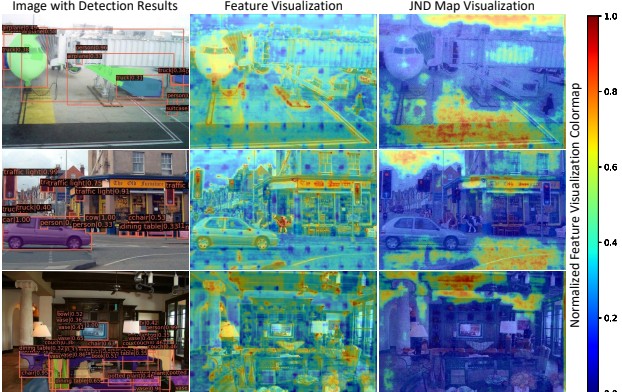

*Figure 9.* Visualization of features and FeatJND maps. It shows FeatJND can preserve crucial semantic information.

tolerance boundary in feature space, and $\alpha$ can serve as a control knob to set the distortion level for downstream uses.

To visualize the spatial distribution of the predicted tolerance, we map the $\ell_2$-norm of FeatJND perturbations using the high-resolution first layer of the FPN based on Mask R-CNN with Swin-T. As shown in Fig. 9, FeatJND effectively suppresses task-irrelevant background regions.

### 4.4. Attribution Map Visualization for FeatJND

To visually investigate the impact of the FeatJND-based distortion on feature maps, we compute the attribution maps for the feature outputs before and after perturbation. The attribution map is designed to highlight which regions of the feature are influential for the classification decision. We adopt an axiomatic attribution method as introduced in (Sundararajan et al., 2017), where the attribution of an input image $I$ is defined as:

$$\text{Attr}(I) = \int_0^1 \frac{\partial \, h_{\text{cls}}\big(\gamma(x)\big)}{\partial \, \gamma(x)} \cdot \frac{\partial \, \gamma(x)}{\partial x} \, dx, \qquad (19)$$

where $I$ is the input image, and $\gamma(x)$ represents an integral path over the input image $I$. For the network without FeatJND distortion, $h_{\text{cls}}$ refers to the classification network output, while for the network with FeatJND-induced distortion, $h_{\text{cls}}$ refers to the classification head applied to the

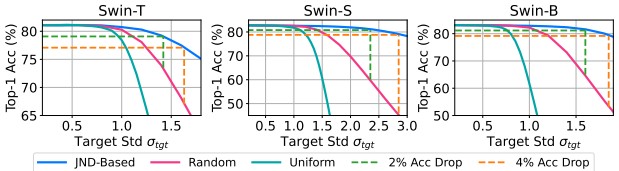

*Figure 10.* Comparison of image classification among JND-based step size, random, and global uniform baselines.

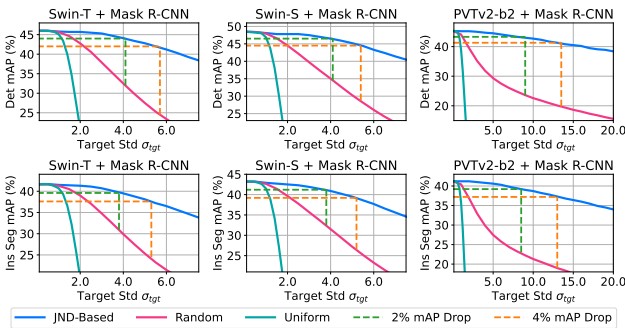

*Figure 11.* Comparison of detection and instance segmentation among JND-based step size, random, and global uniform baselines.

backbone output with the predicted FeatJND map added. In practice, we discretize this integral for implementation purposes, resulting in:

$$\text{Attr}(I) = \sum_{i=1}^{K} \frac{\partial h_{\text{cls}}(I_i)}{\partial(I_i)} \cdot \left( I_i - I_{i-1} \right), \qquad (20)$$

where $I_1, I_2, \ldots, I_K$ represent the discretized path and $K$ is the number of integration steps. For simplicity, we set $I_0$ as a zero matrix and construct the integral path as $I_i = iI/K$.

We conduct a comparison of attribution maps before and after applying FeatJND-based perturbations with $K = 20$. In most cases, the attribution maps for features distorted with FeatJND and the original features *show minimal differences*, confirming that the FeatJND-based distortions are effectively imperceptible to the downstream task. As shown in Fig. 7, after applying FeatJND, some previously highlighted irrelevant areas, which are not important for the final task results, are reduced, indicating that FeatJND effectively removes non-critical parts of the feature, preserving only the most task-relevant information.

## 4.5. Ablation Studies

We analyze the sensitivity of FeatJND to four design choices: the loss-balancing weight $\lambda_{\text{t}}$ in Eq. (8), the temperature $T$ in the task discrepancy term, the number of residual blocks $N_{\text{res}}$ in the FeatJND estimator $G_\theta$, and the spatial kernel size $K_{\text{size}}$ inside the residual blocks. Note that $\tau$ in Eq. (3) only appears as an intermediate quantity and is absorbed into the fixed $\lambda_{\text{t}}$ in the final two-term objective in Eq. (8), so it is not a separately tunable hyperparameter in our implementation. Ablations are conducted on Swin-S for ImageNet-1K classi-

*Table 2.* Ablation studies of FeatJND on Swin-S for ImageNet-1K classification, with Top-1 accuracy reported, and on Swin-S + Mask R-CNN for COCO detection, with mAP@[0.5:0.95] reported, under matched NRMSE. All values are in percentage. Default settings used in the main paper are in **bold**.

| Factor | Value | NRMSE | | | |
|---|---|---|---|---|---|
| | | 10 | 20 | 30 | 40 |
| Classification: Swin-S, Top-1 Acc (%) | | | | | |
| $\lambda_{\text{t}}$ | 10 | 82.0 | 76.0 | 65.7 | 53.7 |
| | 30 | 82.8 | 82.7 | 82.6 | 82.4 |
| | **50** | 82.8 | 82.7 | 82.6 | 82.5 |
| | 70 | 82.8 | 82.7 | 82.6 | 82.5 |
| $T$ | 1 | 82.6 | 82.0 | 80.8 | 78.7 |
| | **4** | 82.8 | 82.7 | 82.6 | 82.5 |
| | 8 | 82.8 | 82.8 | 82.6 | 82.4 |
| $N_{\text{res}}$ | 1 | 82.8 | 82.7 | 82.6 | 82.5 |
| | 2 | 82.8 | 82.7 | 82.6 | 82.5 |
| | **3** | 82.8 | 82.7 | 82.6 | 82.5 |
| | 4 | 82.7 | 82.7 | 82.6 | 82.4 |
| $K_{\text{size}}$ | **1** | 82.8 | 82.7 | 82.6 | 82.5 |
| | 3 | 82.8 | 82.7 | 82.5 | 82.2 |
| | 5 | 82.7 | 82.5 | 82.2 | 81.8 |

| Factor | Value | NRMSE | | | |
|---|---|---|---|---|---|
| | | 2 | 3 | 4 | 5 |
| Detection: Swin-S + Mask R-CNN, mAP (%) | | | | | |
| $\lambda_{\text{t}}$ | 10 | 46.9 | 44.7 | 41.8 | 38.3 |
| | 100 | 48.2 | 47.9 | 47.6 | 47.0 |
| | **200** | 48.4 | 48.2 | 47.9 | 47.5 |
| | 300 | 48.4 | 48.3 | 48.0 | 47.7 |
| $T$ | 1 | 47.7 | 46.9 | 46.0 | 45.1 |
| | **4** | 48.4 | 48.2 | 47.9 | 47.5 |
| | 8 | 48.4 | 48.1 | 47.9 | 47.7 |

fication and on Swin-S + Mask R-CNN for COCO detection, with the default setting highlighted in bold in Table 2.

As shown in Table 2, FeatJND is robust to its main hyperparameters within a broad practical range. When $\lambda_{\text{t}}$ is too small, e.g., $\lambda_{\text{t}} = 10$, the magnitude term dominates and the predicted FeatJND map over-distorts the feature, leading to clear degradation at larger NRMSE. For instance, the Top-1 accuracy drops from $82.0\%$ to $53.7\%$ as NRMSE grows from 10 to 40. Once $\lambda_{\text{t}}$ is in a reasonable range, the performance becomes essentially flat. A larger $T$ yields softer targets and more stable gradients in the KL-based discrepancy term, and the default $T = 4$ already saturates the gain. The depth $N_{\text{res}}$ and kernel size $K_{\text{size}}$ have very small effects, matching our design intuition in Sec. 3.3 that $1 \times 1$ kernels suffice because backbone and neck features have already aggregated spatial context through multiple downsampling stages. Larger kernels such as $K_{\text{size}} = 5$ even slightly hurt performance at high NRMSE.

## 4.6. Feature Quantization as a Proxy Application

To evaluate the benefit of JND-guided resource allocation in approximate settings with limited precision and bandwidth, we treat the intermediate features $f$ as quantizable signals. We apply *token-wise dynamic quantization*, where each to-

ken has a quantization step size $\Delta$, and the same $\Delta$ is used across all channels. The output of the FeatJND model is converted into a non-negative tolerance map $s$, which represents the relative tolerance for each token. This tolerance map $s$ is computed by normalizing the feature change induced by the FeatJND perturbation: $s = |\delta_{\text{JND}}| / (|f| + \epsilon)$ where $\delta_{\text{JND}}$ is the predicted FeatJND map, $f$ is the original feature, and $\epsilon$ is a small constant to avoid division by zero. The token-wise quantization step size map $\Delta$ is then constructed as $\Delta = \lambda s$, where $\lambda$ is a scaling factor to control the quantization distortion. The feature is uniformly quantized based on the token-wise step size map $\Delta$:

$$Q(f) = \Delta \cdot \text{round}(f / \Delta). \tag{21}$$

To ensure a fair comparison with a random baseline, we fix the quantization noise budget by assuming a uniform quantization error variance of $\Delta^2/12$, where the rationale behind this value is provided in the Appendix. We choose $\lambda$ such that $\mathbb{E}(\Delta^2/12) = \sigma_{\text{tgt}}^2$, where $\sigma_{\text{tgt}}$ is the target quantization noise standard deviation per feature element. The random baseline uses the same $s$ distribution but randomly permutes the tokens, ensuring that $\mathbb{E}(s^2)$ remains unchanged, and only the allocation strategy is compared.

We conduct experiments on transformer-based networks for classification, detection, and instance segmentation tasks. We observe that, under the same $\sigma_{\text{tgt}}$, features with JND-based step size allocation outperform the random baseline where step sizes are permuted. For the random baseline, we use 5 random seeds. Besides, both the above methods significantly outperform global uniform step quantization under the same distortion strength. The comparisons are shown in Fig. 10 and Fig. 11, where "Random" denotes the mean over 5 random seeds, and "Uniform" denotes global uniform step size. Since the performance differences across different seeds are small, no shadow areas are plotted around the curves for the variance. Overall, JND-based allocation consistently yields higher task performance under matched quantization strength, validating task-aligned step size allocation under comparable noise budgets.

## 5. Conclusion and Discussions

**Conclusion.** In this work, we take a first step toward a task-aligned *just noticeable difference* for deep visual features by introducing FeatJND. It predicts the maximum tolerable per-feature perturbation while keeping downstream outputs nearly unchanged. We formulate FeatJND as a magnitude-consistency trade-off and evaluate it on classification, detection, and instance segmentation. Across tasks, FeatJND preserves higher performance than Gaussian perturbations at matched NRMSE, and it also improves token-wise feature quantization via JND-guided step-size allocation under the same noise budget. Overall, FeatJND offers a practical way to characterize and control deep feature quality through a

task-aligned tolerance boundary. FeatJND may serve as a general building block for feature-centric systems by providing a task-aligned tolerance boundary to characterize and control feature quality, and could be extended to support more downstream tasks.

**Discussions.** This work is an initial step toward feature JND, and several directions remain open. Promising future work includes developing a task-/model-general FeatJND predictor that transfers across objectives; exploring structured noise patterns for high-efficiency, such as low-rank structure in large-model features; and integrating FeatJND into downstream applications such as visual feature compression (Gao et al., 2025b) and feature watermarking (Rezaei et al., 2024) to enable more effective distortion shaping.

## Impact Statement

This paper presents work whose goal is to advance the field of machine learning. There are many potential societal consequences of our work, none of which we feel must be specifically highlighted here.

## Acknowledgements

This work was supported by the Ministry of Education of Singapore under Grant T2EP20123-0006.

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

## A. NRMSE vs Cosine Similarity as Distortion Budget

Cosine similarity is a widely used metric for measuring the similarity between two feature vectors. However, in this work we need a scalar axis that can serve as a consistent distortion budget to match perturbation strength across models and methods. We therefore quantify distortion by the normalized root mean squared error:

$$\text{NRMSE}(f, \tilde{f}) \triangleq \frac{\|\tilde{f} - f\|_2}{\|f\|_2 + \epsilon} = \frac{\|e\|_2}{\|f\|_2 + \epsilon}, \qquad \tilde{f} = f + e. \tag{22}$$

Cosine similarity is defined as

$$\cos(f, \tilde{f}) \triangleq \frac{\langle f, \tilde{f} \rangle}{\|f\|_2 \|\tilde{f}\|_2}. \tag{23}$$

We use cosine similarity only as an auxiliary descriptor of geometric alignment, not as the main distortion-strength axis.

### A.1. Insensitivity to Rescaling

For a pure rescaling perturbation $e = (\gamma - 1)f$ so that $\tilde{f} = \gamma f$ with $\gamma > 0$, we have

$$\cos(f, \tilde{f}) = 1 \qquad \text{but} \qquad \text{NRMSE}(f, \tilde{f}) \approx |\gamma - 1|. \tag{24}$$

Hence cosine similarity may indicate no change even when the relative error magnitude becomes large, so it cannot serve as a reliable distortion budget axis.

### A.2. Saturation under Large Distortions

Let $r$ denote the relative distortion magnitude

$$r \triangleq \frac{\|e\|_2}{\|f\|_2}. \tag{25}$$

A representative and common case is that the perturbation is approximately orthogonal to the feature, which is a good approximation for many zero-mean isotropic noises at sufficiently high dimension. When $\langle f, e \rangle \approx 0$, we obtain

$$\cos(f, \tilde{f}) = \frac{\|f\|_2}{\|f + e\|_2} \approx \frac{1}{\sqrt{1 + r^2}} = \frac{1}{\sqrt{1 + \text{NRMSE}^2}}. \tag{26}$$

This mapping quickly saturates when $r$ is large. Its sensitivity decays as

$$\left| \frac{\mathrm{d}}{\mathrm{d}r} \frac{1}{\sqrt{1 + r^2}} \right| = \frac{r}{(1 + r^2)^{3/2}} \sim \frac{1}{r^2} \qquad \text{as} \qquad r \to \infty. \tag{27}$$

Therefore, in the large-distortion regime, even a substantial change in distortion magnitude produces only a small change in cosine similarity, making it difficult to distinguish different distortion levels using cosine similarity. This is consistent with the empirical NRMSE–CosSim curves in Fig. 12, where cosine similarity drops rapidly and then becomes much less discriminative as distortion increases.

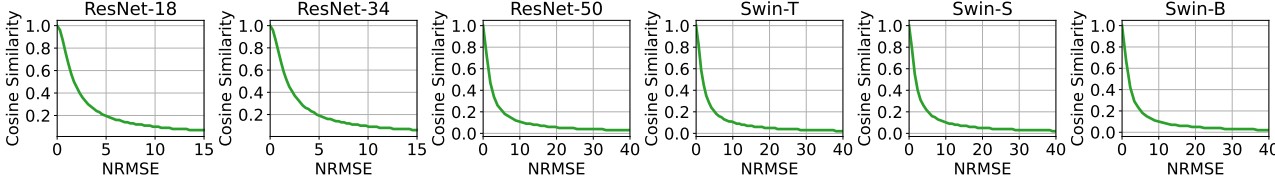

*Figure 12.* Cosine similarity versus feature distortion measured by NRMSE on ImageNet-1K classification across different backbones.

### A.3. Inconsistent Budget across Distortion Structures

A budget metric should provide sufficient resolution to distinguish performance under different distortion types at comparable distortion strengths. Fig. 13 shows that for strong backbones such as ResNet-50 and Swin variants, the Top-1 accuracy quickly saturates once cosine similarity exceeds a small threshold, and over the vast majority of the cosine-similarity range the FeatJND and Gaussian curves are nearly indistinguishable. This indicates cosine similarity has limited discriminability as a distortion-strength axis for comparing methods, especially in the practically relevant low-to-moderate distortion regime. In contrast, NRMSE directly measures relative distortion magnitude and offers a more stable and interpretable budget for matching distortion strength across methods and models. In other words, cosine similarity leaves only a narrow transition region where accuracy changes noticeably, making it a less suitable budget for systematic comparisons.

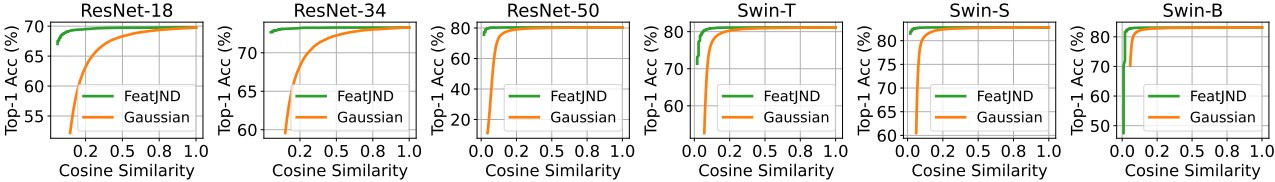

*Figure 13.* Top-1 accuracy versus cosine similarity under FeatJND and Gaussian feature distortion on ImageNet-1K classification.

### A.4. NRMSE vs NMSE

Both NRMSE and NMSE are reasonable normalized distortion measures, but they quantify different quantities. NRMSE measures the relative error magnitude, while NMSE measures the relative error energy

$$\text{NMSE}(f, \tilde{f}) \triangleq \frac{\|\tilde{f} - f\|_2^2}{\|f\|_2^2 + \epsilon}. \tag{28}$$

At the sample level, NMSE is approximately the squared NRMSE when the same normalization is used, so they induce the same ordering over distortion strengths. We report NRMSE because our implementation computes the normalized $\ell_2$ magnitude ratio in Eq. (22), and NRMSE is more directly interpretable as a relative magnitude.

## B. Deriving the Budget under Token-Wise Uniform Quantization

We derive why the budget term $\mathbb{E}(\Delta^2/12)$ naturally arises from the standard *uniform quantization error model*. The key idea is that, under mild conditions, the quantization error within one quantization bin can be approximated as uniformly distributed, yielding a variance of $\Delta^2/12$. When the step size $\Delta$ varies across tokens, averaging the per-token variance leads to the global budget $\mathbb{E}(\Delta^2/12)$.

**(1) Scalar uniform quantization.** Consider a scalar $x$ and the round-to-nearest uniform quantizer with step size $\Delta$:

$$Q(x) = \Delta \cdot \text{round}(x/\Delta). \tag{29}$$

Define the quantization error

$$\epsilon = Q(x) - x. \tag{30}$$

Write $x$ as an integer grid point plus a residual:

$$x = k\Delta + r, \qquad r \in \left[-\frac{\Delta}{2}, \frac{\Delta}{2}\right), \quad k \in \mathbb{Z}. \tag{31}$$

Since $Q(x)$ maps $x$ to the nearest grid point $k\Delta$, we have

$$Q(x) = k\Delta, \qquad \epsilon = k\Delta - (k\Delta + r) = -r, \tag{32}$$

and therefore

$$\epsilon \in \left[-\frac{\Delta}{2}, \frac{\Delta}{2}\right). \tag{33}$$

**(2) Why a uniform distribution approximation is reasonable.** A classical quantization-noise assumption in signal processing and compression is that, if the signal is sufficiently smooth within each quantization bin, then the residual $r$ is approximately uniform over $[-\Delta/2, \Delta/2)$. Hence,

$$\epsilon \sim \mathcal{U}\left(-\frac{\Delta}{2}, \frac{\Delta}{2}\right), \tag{34}$$

which is an approximation but widely used as a proxy for budget control.

**(3) Variance of a uniform random variable:** $\mathrm{Var}(\epsilon) = \Delta^2/12$. Let $\epsilon \sim \mathcal{U}(-a, a)$ with $a = \Delta/2$. Its density is

$$f(\epsilon) = \frac{1}{2a}, \qquad \epsilon \in [-a, a]. \tag{35}$$

By symmetry, $\mathbb{E}(\epsilon) = 0$. The second moment is

$$\mathbb{E}(\epsilon^2) = \int_{-a}^{a} \epsilon^2 \cdot \frac{1}{2a} \, d\epsilon = \frac{1}{2a}\left(\frac{\epsilon^3}{3}\right)\Big|_{-a}^{a} = \frac{1}{2a} \cdot \frac{2a^3}{3} = \frac{a^2}{3}. \tag{36}$$

Therefore,

$$\mathrm{Var}(\epsilon) = \mathbb{E}(\epsilon^2) - \mathbb{E}(\epsilon)^2 = \frac{a^2}{3} = \frac{(\Delta/2)^2}{3} = \frac{\Delta^2}{12}. \tag{37}$$

**(4) From token-wise step sizes to the global budget** $\mathbb{E}(\Delta^2/12)$. In token-wise quantization, the step size is a *token-level step map* shared across channels. For a CNN feature $z \in \mathbb{R}^{C \times H \times W}$, where we omit the batch index for simplicity. We use one step per spatial token:

$$\Delta_{h,w} \quad \text{shared across all channels } c. \tag{38}$$

Applying the same uniform-noise model per element yields

$$\mathrm{Var}(\epsilon_{c,h,w}) \approx \frac{\Delta_{h,w}^2}{12}. \tag{39}$$

To control the overall quantization strength with a single scalar, we average the per-element noise energy, giving

$$\underbrace{\mathbb{E}(\epsilon^2)}_{\text{average noise energy}} \approx \mathbb{E}\left(\frac{\Delta_{h,w}^2}{12}\right), \tag{40}$$

where the expectation is taken over all elements $(c, h, w)$. Since $\Delta$ is shared across the channel dimension $c$, the same expectation can be equivalently viewed as an average over spatial tokens $(h, w)$. We therefore define the noise budget via

$$\sigma_{\text{std}}^2 \triangleq \mathbb{E}\left(\frac{\Delta_{h,w}^2}{12}\right). \tag{41}$$

**(5) Solving for the scaling factor** $\lambda$ **when** $\Delta = \lambda s$. When we parameterize the token-wise step map as $\Delta = \lambda s$, where $s \geq 0$ denotes a token-level tolerance map, the budget constraint becomes

$$\mathbb{E}\left(\frac{(\lambda s)^2}{12}\right) = \sigma_{\text{std}}^2 \implies \lambda^2 \cdot \frac{\mathbb{E}(s^2)}{12} = \sigma_{\text{std}}^2 \implies \lambda = \sqrt{\frac{12 \cdot \sigma_{\text{std}}^2}{\mathbb{E}(s^2)}}. \tag{42}$$

## C. Additional Results with Extended COCO Metrics

### C.1. COCO Metrics Used in Detection and Instance Segmentation

We use the standard COCO metrics for object detection with box IoU and instance segmentation with mask IoU. For an IoU threshold $t$, a prediction is a true positive if its IoU with a ground-truth instance is at least $t$. **AP@$t$** is the average precision at threshold $t$ averaged over categories. **AP@50** uses $t = 0.50$ and is more lenient, while **AP@75** uses $t = 0.75$ and is stricter and more sensitive to localization. The main metric **mAP** is **mAP@[0.5:0.95]**, which averages AP over thresholds from $0.50$ to $0.95$ with step $0.05$. We also report **mAR@[0.5:0.95]**, which averages recall over the same threshold range under the COCO protocol.

## C.2. Extended Results for Gradually Increasing FeatJND Strength

Fig. 14 and Fig. 15 extend Fig. 8 by reporting detection and instance-segmentation performance drops under additional COCO metrics as the FeatJND weight $\alpha$ increases. Besides the main mAP@[0.5:0.95], we include AP@50, AP@75, and mAR@[0.5:0.95] to provide a finer-grained view: AP@50 emphasizes coarse correctness, AP@75 stresses accurate localization, and mAR summarizes recall behavior across IoU thresholds. These extended curves verify that the same "gradually adding JND" behavior observed in Fig. 8 is consistent across both precision- and recall-oriented metrics.

## C.3. Extended Comparisons under Matched Distortion Strength

Fig. 16 and Fig. 17 extend Fig. 5 and Fig. 6 by evaluating FeatJND-based distortions versus Gaussian perturbations under the same NRMSE but with additional COCO metrics. Reporting AP@50, AP@75, and mAR@[0.5:0.95] complements mAP@[0.5:0.95] and helps disentangle whether the gains mainly come from improved localization precision (AP@75) or improved coverage/recall (mAR).

## C.4. Extended Quantization Application Results

Fig. 18 and Fig. 19 extend Fig. 11 by reporting more COCO metrics for the token-wise quantization proxy application. Under the same quantization noise budget, we compare FeatJND-guided step-size allocation with the random permutation baseline and the global uniform baseline using AP@50, AP@75, and mAR@[0.5:0.95] in addition to the main mAP@[0.5:0.95]. The consistent improvements across these metrics further support that FeatJND provides task-aligned tolerances that are beneficial for resource allocation.

Table 3 provides a comprehensive numerical summary of the performance comparison between the proposed FeatJND-based quantization and the random step-size permutation baseline. We report the exact performance values and the corresponding positive gains ($\Delta$) for ImageNet classification (Top-1 Acc), COCO object detection (mAP), and instance segmentation (mAP) across various Transformer-based backbones, including Swin-T/S/B and PVTv2-b1/b2/b3. Consistent with the visual evidence in Fig. 10 and Fig. 11, the tabular results confirm that under a matched noise budget, prioritizing token precision based on FeatJND predictions yields consistently superior task performance compared to random allocation across all evaluated model variants and tasks.

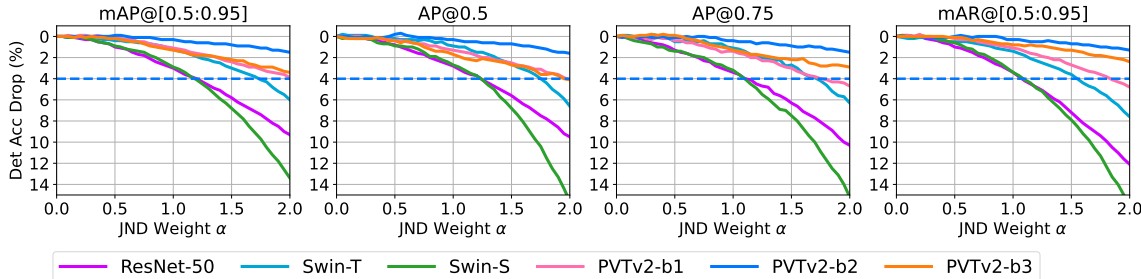

*Figure 14.* Detection results with additional COCO metrics under gradually increased FeatJND strength, extending Fig. 8.

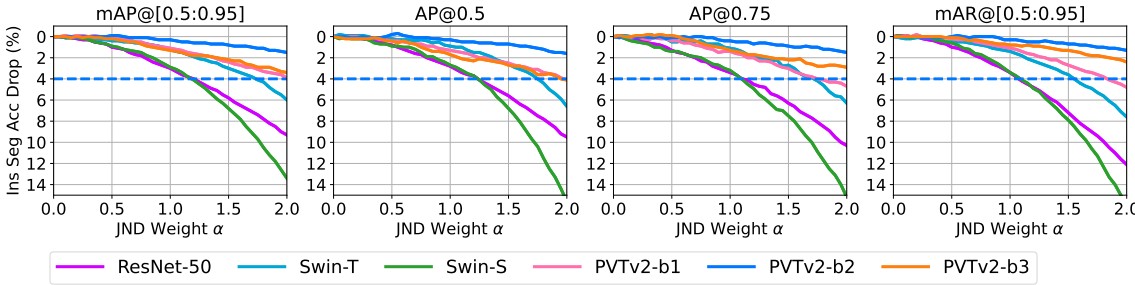

*Figure 15.* Instance segmentation results with additional COCO metrics under gradually increased FeatJND strength, extending Fig. 8.

*Table 3.* Performance comparison (%) of FeatJND-based token-wise quantization over the random step-size permutation baseline.

| Task | $\sigma_{std}$ | Method | Swin | | | PVTv2 | | |
|---|---|---|---|---|---|---|---|---|
| | | | –T | –S | –B | –b1 | –b2 | –b3 |
| Classification | 1.6 | FeatJND-Based | 77.5 | 82.4 | 81.6 | – | – | – |
| | | Random | 68.1 | $78.5_{\pm 0.12}$ | $64.6_{\pm 0.095}$ | – | – | – |
| | | $\Delta$ | 9.37 | $3.83_{\pm 0.12}$ | $17.0_{\pm 0.095}$ | – | – | – |
| | 1.8 | FeatJND-Based | 75.1 | 82.2 | 80.1 | – | – | – |
| | | Random | 61.7 | $74.7_{\pm 0.077}$ | $55.5_{\pm 0.086}$ | – | – | – |
| | | $\Delta$ | 13.4 | $7.50_{\pm 0.077}$ | $24.5_{\pm 0.086}$ | – | – | – |
| | 2.0 | FeatJND-Based | 72.1 | 81.9 | 77.6 | – | – | – |
| | | Random | 55.3 | $69.7_{\pm 0.057}$ | $47.1_{\pm 0.077}$ | – | – | – |
| | | $\Delta$ | 16.8 | $12.2_{\pm 0.057}$ | $30.6_{\pm 0.077}$ | – | – | – |
| | 2.2 | FeatJND-Based | 68.7 | 81.6 | 74.1 | – | – | – |
| | | Random | 49.4 | $64.1_{\pm 0.16}$ | $39.6_{\pm 0.027}$ | – | – | – |
| | | $\Delta$ | 19.3 | $17.5_{\pm 0.16}$ | $34.4_{\pm 0.027}$ | – | – | – |
| | 2.4 | FeatJND-Based | 65.0 | 81.0 | 69.9 | – | – | – |
| | | Random | 43.9 | $58.3_{\pm 0.16}$ | $33.4_{\pm 0.11}$ | – | – | – |
| | | $\Delta$ | 21.1 | $22.8_{\pm 0.16}$ | $36.6_{\pm 0.11}$ | – | – | – |
| Detection | 2.0 | FeatJND-Based | 45.7 | 47.8 | – | 41.6 | 45.2 | 46.8 |
| | | Random | 42.8 | $44.5_{\pm 0.040}$ | – | $37.9_{\pm 0.080}$ | $39.5_{\pm 0.10}$ | $37.6_{\pm 0.049}$ |
| | | $\Delta$ | 2.94 | $3.28_{\pm 0.040}$ | – | $3.66_{\pm 0.080}$ | $5.74_{\pm 0.10}$ | $9.24_{\pm 0.049}$ |
| | 4.0 | FeatJND-Based | 44.1 | 46.5 | – | 40.8 | 44.7 | 46.5 |
| | | Random | 32.5 | $35.0_{\pm 0.075}$ | – | $30.2_{\pm 0.080}$ | $31.8_{\pm 0.040}$ | $31.8_{\pm 0.080}$ |
| | | $\Delta$ | 11.6 | $11.5_{\pm 0.075}$ | – | $10.6_{\pm 0.080}$ | $12.9_{\pm 0.040}$ | $14.7_{\pm 0.080}$ |
| | 6.0 | FeatJND-Based | 41.2 | 43.3 | – | 39.8 | 44.1 | 46.1 |
| | | Random | 23.4 | $26.1_{\pm 0.10}$ | – | $24.5_{\pm 0.080}$ | $27.5_{\pm 0.075}$ | $29.0_{\pm 0.12}$ |
| | | $\Delta$ | 17.8 | $17.2_{\pm 0.10}$ | – | $15.3_{\pm 0.080}$ | $16.6_{\pm 0.075}$ | $17.1_{\pm 0.12}$ |
| | 8.0 | FeatJND-Based | 37.5 | 39.4 | – | 38.5 | 43.3 | 45.8 |
| | | Random | 17.5 | $19.5_{\pm 0.10}$ | – | $20.2_{\pm 0.049}$ | $24.8_{\pm 0.10}$ | $27.1_{\pm 0.080}$ |
| | | $\Delta$ | 20.0 | $19.9_{\pm 0.10}$ | – | $18.3_{\pm 0.049}$ | $18.5_{\pm 0.10}$ | $18.7_{\pm 0.080}$ |
| | 10.0 | FeatJND-Based | 33.7 | 35.1 | – | 36.7 | 42.7 | 45.4 |
| | | Random | 13.9 | $15.0_{\pm 0.080}$ | – | $17.1_{\pm 0.13}$ | $22.6_{\pm 0.075}$ | $25.6_{\pm 0.063}$ |
| | | $\Delta$ | 19.8 | $20.1_{\pm 0.080}$ | – | $19.6_{\pm 0.13}$ | $20.1_{\pm 0.075}$ | $19.8_{\pm 0.063}$ |
| Instance Segmentation | 2.0 | FeatJND-Based | 41.3 | 42.5 | – | 38.6 | 41.2 | 42.4 |
| | | Random | 39.0 | $39.7_{\pm 0.049}$ | – | $35.4_{\pm 0.075}$ | $36.8_{\pm 0.049}$ | $34.6_{\pm 0.075}$ |
| | | $\Delta$ | 2.32 | $2.84_{\pm 0.049}$ | – | $3.18_{\pm 0.075}$ | $4.36_{\pm 0.049}$ | $7.82_{\pm 0.075}$ |
| | 4.0 | FeatJND-Based | 39.5 | 40.9 | – | 37.7 | 40.8 | 42.1 |
| | | Random | 29.9 | $31.5_{\pm 0.063}$ | – | $28.3_{\pm 0.040}$ | $29.8_{\pm 0.080}$ | $28.8_{\pm 0.040}$ |
| | | $\Delta$ | 9.60 | $9.40_{\pm 0.063}$ | – | $9.38_{\pm 0.040}$ | $11.0_{\pm 0.080}$ | $13.3_{\pm 0.040}$ |
| | 6.0 | FeatJND-Based | 36.5 | 37.7 | – | 36.3 | 40.3 | 41.7 |
| | | Random | 21.5 | $23.3_{\pm 0.049}$ | – | $22.9_{\pm 0.049}$ | $25.8_{\pm 0.049}$ | $25.9_{\pm 0.12}$ |
| | | $\Delta$ | 15.0 | $14.4_{\pm 0.049}$ | – | $13.4_{\pm 0.049}$ | $14.5_{\pm 0.049}$ | $15.8_{\pm 0.12}$ |
| | 8.0 | FeatJND-Based | 32.9 | 33.6 | – | 34.7 | 39.4 | 41.3 |
| | | Random | 16.1 | $17.1_{\pm 0.049}$ | – | $19.0_{\pm 0.049}$ | $23.2_{\pm 0.10}$ | $24.0_{\pm 0.089}$ |
| | | $\Delta$ | 16.8 | $16.5_{\pm 0.049}$ | – | $15.7_{\pm 0.049}$ | $16.2_{\pm 0.10}$ | $17.3_{\pm 0.089}$ |
| | 10.0 | FeatJND-Based | 29.3 | 29.4 | – | 32.8 | 38.7 | 41.0 |
| | | Random | 12.9 | $13.0_{\pm 0.075}$ | – | $16.2_{\pm 0.080}$ | $21.3_{\pm 0.063}$ | $22.6_{\pm 0.075}$ |
| | | $\Delta$ | 16.4 | $16.4_{\pm 0.075}$ | – | $16.6_{\pm 0.080}$ | $17.4_{\pm 0.063}$ | $18.4_{\pm 0.075}$ |

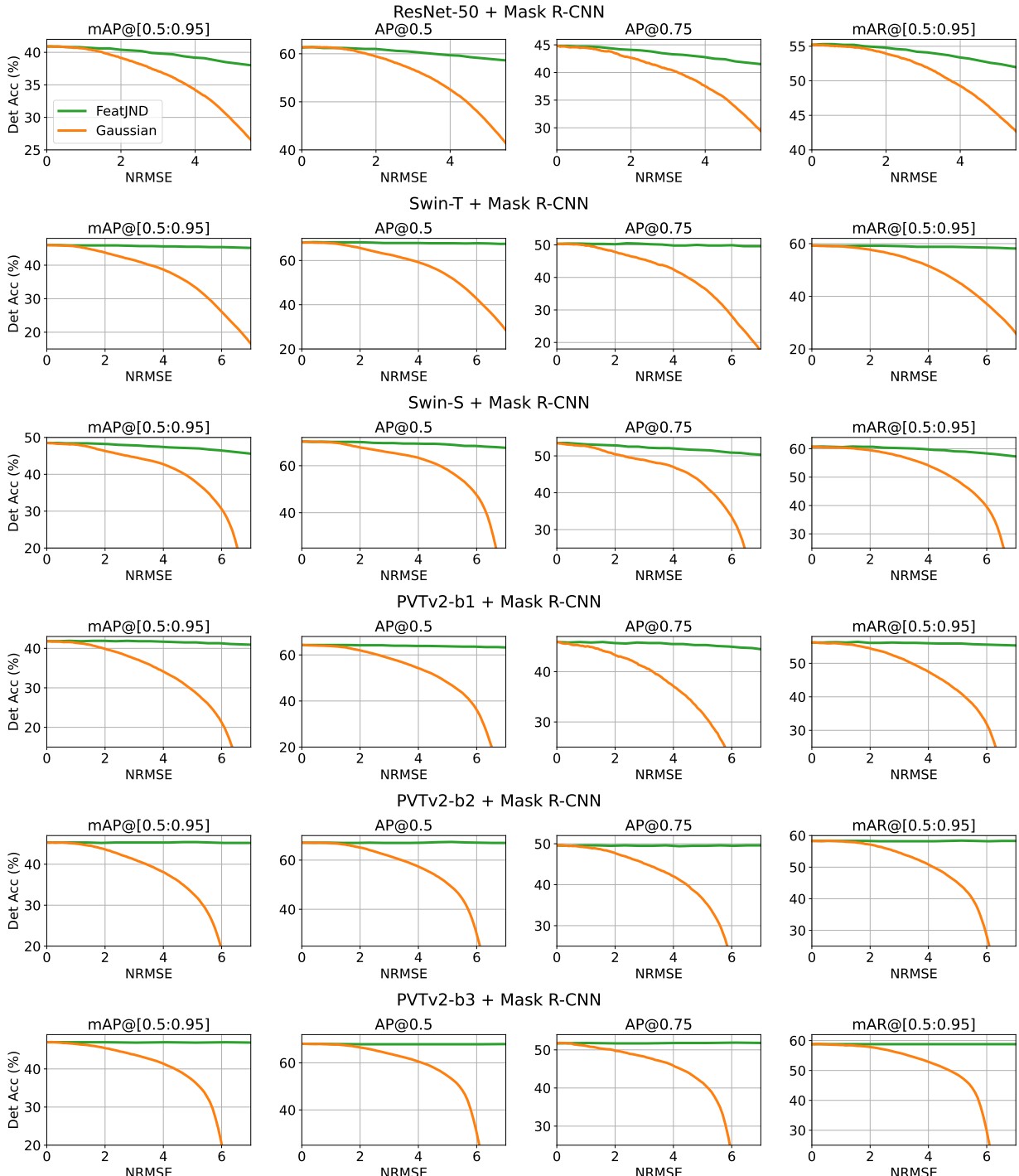

*Figure 16.* Detection results with additional COCO metrics under matched NRMSE for FeatJND and Gaussian, extending Fig. 5.

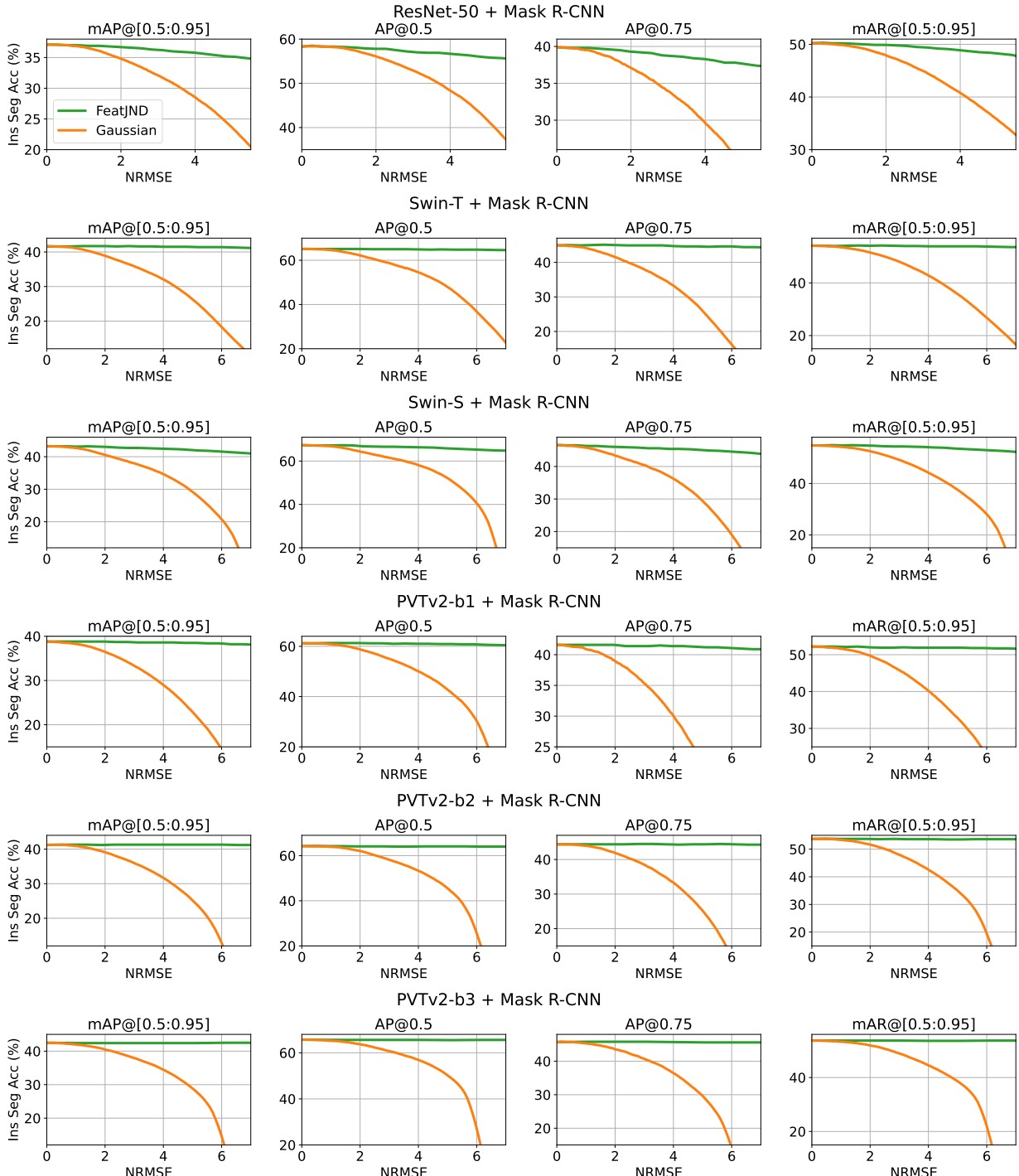

*Figure 17.* Instance segmentation results with additional COCO metrics under matched NRMSE for FeatJND and Gaussian, extending Fig. 6.

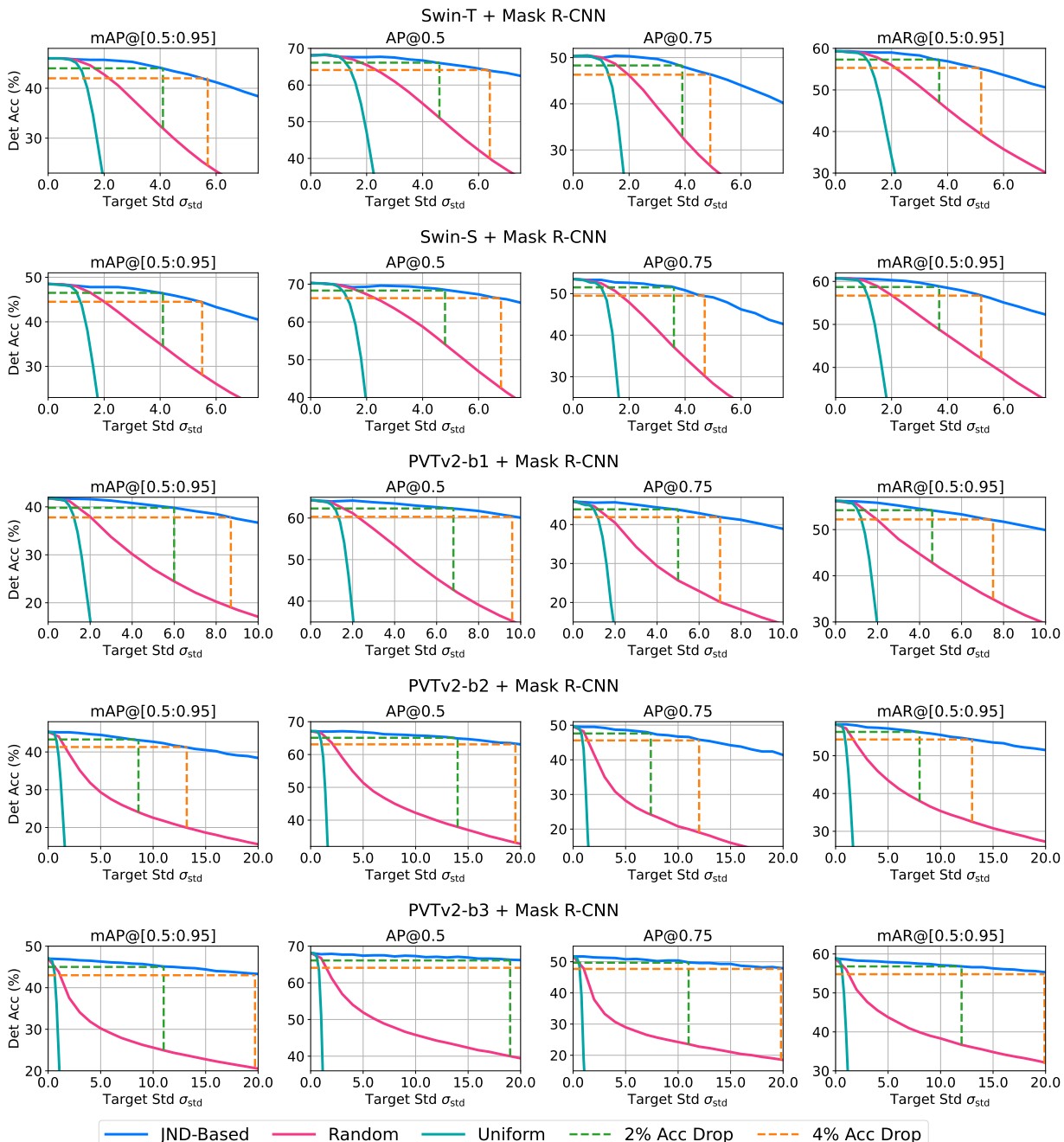

*Figure 18.* Detection quantization results with additional COCO metrics under the same noise budget, extending Fig. 11.

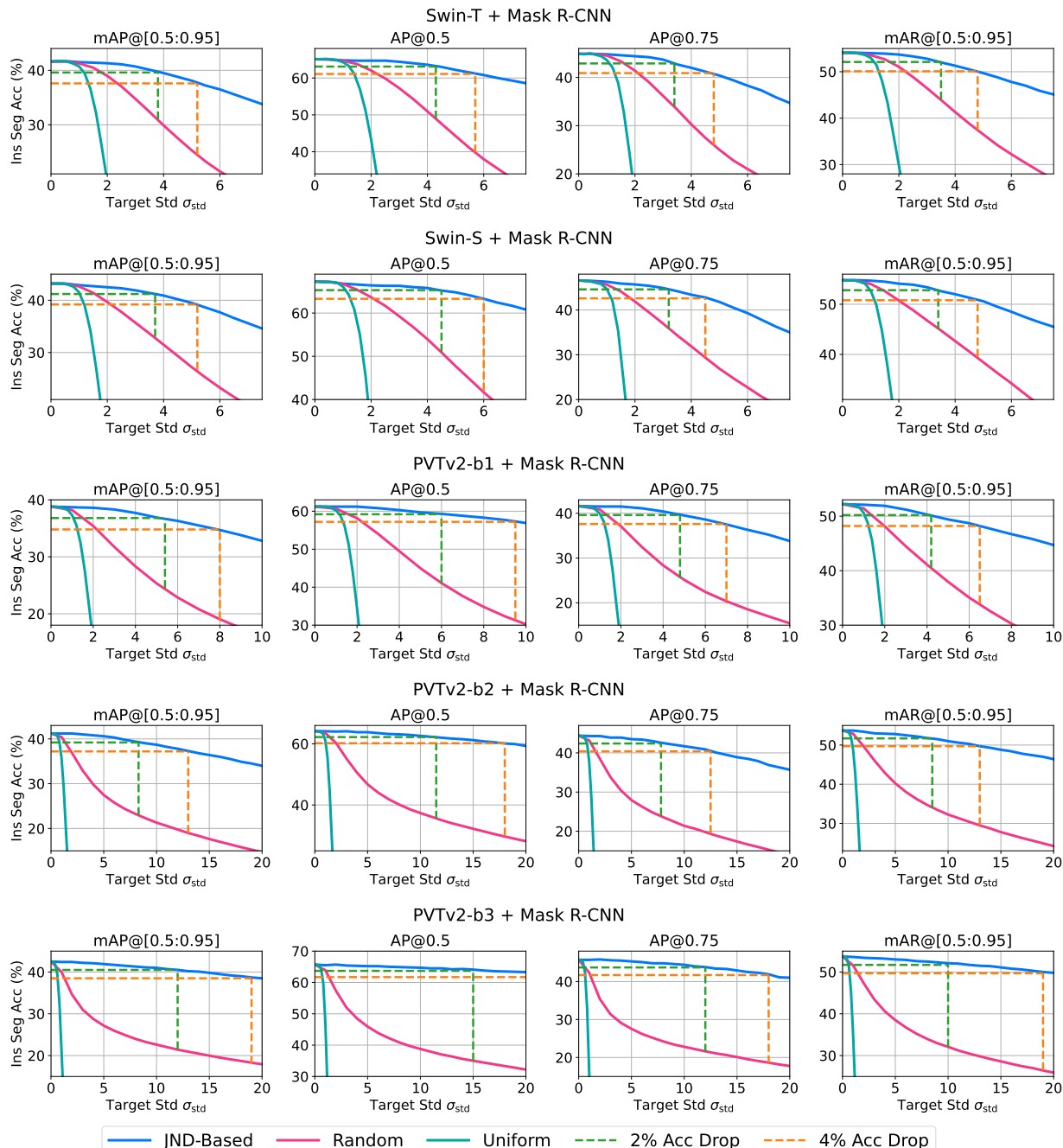

*Figure 19.* Instance segmentation quantization results with additional COCO metrics under the same noise budget, extending Fig. 11.

