# OpenReview forum: "Just Noticeable Difference Modeling for Deep Visual Features"
_ICML.cc/2026/Conference — ICML 2026 spotlight_

### Official Review · Reviewer_K9NV · 2026-02-23

**Soundness:** 4
**Presentation:** 4
**Significance:** 4
**Originality:** 4
**Overall Recommendation:** 5
**Confidence:** 5

**Summary:**

This paper proposes FeatJND for deep visual features motivated by the traditional just noticeable difference (JND) for images. The authors first analyze what properties a JND map should satisfy in deep feature space, and then formulate FeatJND as an intermediate-feature distortion that is as large as possible in magnitude while keeping the downstream task performance nearly unchanged. Based on this formulation, the authors design a simple but effective FeatJND estimator and evaluate it on image classification, object detection, and instance segmentation. The results suggest that, under the same distortion magnitude, FeatJND leads to only minor changes in task performance. Visualizations further indicate that FeatJND tends to concentrate on regions that are less relevant to the final predictions. In addition, the paper presents a token-wise feature quantization study guided by FeatJND, showing that FeatJND-based quantization performs much better than baselines such as uniform step sizes and randomly shuffled step sizes.

**Compliance With Llm Reviewing Policy:**

Affirmed.

**Final Justification:**

After a thorough response, I have no further concerns about the article, and I will retain my score.

**Key Questions For Authors:**

The key questions are listed in the Weaknesses part, including questions related to NRMSE, λ, feature quantization experiments, and data access for the FeatJND.

**Limitations:**

Yes.

**Strengths And Weaknesses:**

*Strengths*

(1) Soundness: The proposed FeatJND is sound both theoretically and in implementation. The authors derive the desired properties of FeatJND from a theoretical perspective and design a comprehensive set of experiments to demonstrate its effectiveness across different tasks and architectures, as well as in downstream applications built on FeatJND.

(2) Presentation: This paper is clearly written and well structured. Moreover, given the novelty of the topic, the related work section clearly explains how it connects to and differs from existing lines of research such as feature quality modeling and adversarial attacks.

(3) Significance: The topic is relevant to modern machine learning based on deep visual features. By introducing FeatJND as a way to characterize feature-space properties, the paper has the potential to benefit future work in areas such as efficient learning and practical deployment of machine learning systems.

(4) Originality: The idea is novel. This is a pioneering work to apply the JND concept to deep features, offering the machine learning community a new perspective for analyzing feature representations.


*Weaknesses*

(1) In the experimental part, why is NRMSE used to measure the distortion level of features? It would be better to provide more explanation.

(2) During training, is λ kept fixed, or does it change over the process of training?

(3) In the feature quantization experiments, why does the term Δ²/12 appear? How is this value derived, or how is it set in practice?

(4) It would be helpful to clarify what information the FeatJND estimator needs to access from the task model and the intermediate features, such as feature values, model parameters, or gradients. This would make the proposed method clearer to readers.

---

> ### Author Rebuttal · Authors · 2026-03-30
>
> We sincerely thank the reviewer for the thorough evaluation and the strong recognition of our theoretical soundness, experimental design, and originality. We are glad that the reviewer considers FeatJND a pioneering and significant contribution. We address each question below.
>
> ### **Q1. Why NRMSE is used to measure feature distortion?**
>
> We use NRMSE because it serves as a consistent and interpretable scalar axis for matching distortion strength across different models and methods. We also considered cosine similarity as an alternative but found it unsuitable for three reasons. (1) Cosine similarity can remain at 1 under pure rescaling perturbations even when the actual distortion magnitude is large. (2) Under large distortions, cosine similarity saturates and loses resolution to distinguish different distortion levels. (3) As shown in Fig. 13 in the Appendix, when plotted against cosine similarity, the FeatJND and Gaussian curves become nearly indistinguishable over the majority of the range, making it a poor budget axis for method comparison. In contrast, NRMSE directly measures relative distortion magnitude and provides stable, discriminative comparisons across models and backbones. A detailed analysis with derivations and empirical curves is provided in Sec. A in the Appendix.
>
>
> ---
> ### **Q2. Is $\lambda$ fixed or changed during training?**
>
> In our implementation, $\lambda$ is kept fixed throughout training as the hyperparameter $\lambda_t$ in Eq. (8). Specifically, we set $\lambda_t = 50$ for classification and $\lambda_t = 200$ for detection and instance segmentation, chosen to balance the task consistency term and the magnitude maximization term to comparable scales.
>
>
> ---
> ### **Q3. Why $\Delta^{2}/12$ appears in the feature quantization experiments?**
>
> The term $\Delta^{2}/12$ arises from the standard uniform quantization error model in signal processing. For a round-to-nearest uniform quantizer with step size $\Delta$, the quantization error $\epsilon = Q(x) - x$ lies in $[-\Delta/2, \Delta/2)$. Under the classical assumption that the signal is sufficiently smooth within each quantization bin, the error is approximately uniformly distributed: $\epsilon \sim U(-\Delta/2, \Delta/2)$. The variance of this uniform distribution is $\text{Var}(\epsilon) = \Delta^{2}/12$. In token-wise quantization where the step size $\Delta_{h,w}$ varies across spatial tokens but is shared across channels, the average quantization noise energy becomes $\mathbb{E}(\Delta_{h,w}^{2}/12)$. We set this equal to a target $\sigma_{\text{tgt}}^{2}$ to control the overall quantization strength with a single scalar, ensuring fair comparison across methods under the same noise budget. The detailed derivation is provided in Sec. B in the Appendix.
>
>
> ---
> ### **Q4. What information the FeatJND estimator accesses?**
>
> We summarize the information access of the FeatJND estimator in the following table.
>
> | Information | Feature values | Model parameters | Model gradients | Ground Truths |
> | ----------- | -------------- | ---------------- | --------------- | ------------- |
> | Training    | Yes            | Yes              | Yes             | No            |
> | Inference   | Yes            | No               | No              | No            |
>
> During training, the estimator $G_{\theta}$ takes feature values as input, and gradients flow through the frozen task head to update $G_{\theta}$'s parameters via the task discrepancy loss. Ground-truth labels are never used; instead, the clean feature's own task outputs serve as the consistency target. At inference, $G_{\theta}$ operates as a purely feed-forward, gradient-free module: it takes only the intermediate feature tensor $f$ as input and directly outputs the FeatJND tolerance map $\delta_{\text{JND}} = G_{\theta}(f)$, without requiring access to any model parameters, gradients, or labels. This makes FeatJND practical for deployment scenarios where the downstream model internals are unavailable.

---

> > ### Author Rebuttal · Reviewer_K9NV · 2026-04-01
> >
> > The authors have addressed my questions. I also reviewed the responses to other reviewers and found them convincing. The ablation studies and direct comparisons with gradient-magnitude maps and Jin et al. (Reviewer Uimk) demonstrate the non-triviality of the feature-domain extension, and the double-noise experiments (Reviewer LEhk) confirm robustness under compound corruption. I maintain my recommendation of Accept.

---

### Official Review · Reviewer_Zm5r · 2026-03-12

**Soundness:** 4
**Presentation:** 4
**Significance:** 4
**Originality:** 4
**Overall Recommendation:** 6
**Confidence:** 5

**Summary:**

This paper introduces FeatJND, a task-aligned notion of “just noticeable difference” in deep visual feature space for feature-centric systems. The core idea is to define a per-feature tolerance map that specifies the maximum perturbation each feature component can tolerate while keeping downstream task outputs nearly unchanged. Besides, this paper also proposes to learn a lightweight estimator that predicts this tolerance map directly from the intermediate feature tensors, without requiring access to model internals during inference. The predicted tolerance boundary can then be used as a practical tool for feature quality control and resource allocation. In the experimental part, this paper shows comparisons between FeatJND-based and Gaussian-based feature distortion, as well as the attribution map for FeatJND, demonstrating the effectiveness of the JND. The paper also applies FeatJND to token-wise feature quantization, further showing the usability of the proposed FeatJND.

**Compliance With Llm Reviewing Policy:**

Affirmed.

**Final Justification:**

Thanks the authors for their responses. All my questions have been addressed. I have no further questions and maintain the positive assessment.

**Key Questions For Authors:**

As shown in the aforementioned Weaknesses part, there are some unclear points. If these points are clarified, the paper will be improved.

**Limitations:**

yes

**Strengths And Weaknesses:**

Strength:

1.This paper transfers the long-established JND concept from image processing to feature-centric machine learning, and supports the soundness of JND in feature space through theoretical modeling, experimental analysis, and visualizations.
2.The proposed approach provides a valuable and novel perspective for the machine learning community, with meaningful contributions to areas such as feature analysis and quality control.
3.This paper is well organized with a clear and complete logical chain. Fig. 1 and Sec. 1 offer a helpful high-level roadmap that makes the overall idea easier to follow.
4.This paper evaluates distortion handling and feature quantization across multiple tasks. For each task, the paper uses different architectures, which helps demonstrate the robustness of the proposed method.


Weaknesses:

There are some unclear points that need clarification.
1.Compared with the downstream task model, how about the inference time of the proposed JND estimator?
2.In Sec. 4.3, what is the role of the parameter $\alpha$?
3.$\alpha$ seems to be related to specific applications of FeatJND. What values of $\alpha$ are used in Fig. 7 and Fig. 9?
4.In the feature quantization setting, how is the step-size map obtained in detail?

---

> ### Author Rebuttal · Authors · 2026-03-30
>
> We sincerely thank the reviewer for the thorough evaluation and the strong recognition of our formulation, experimental design, and practical contributions. We are glad that the reviewer finds FeatJND to be a valuable and novel perspective for the community. We address the clarification questions below.
>
> ### **Q1. Inference time of the JND estimator compared with the downstream task model**
>
> The following shows the single-frame inference latency on an NVIDIA 4090 GPU for the ImageNet classification task. Across these backbones, **the JND estimator has a nearly constant latency of approximately *0.8ms***, which is lightweight relative to the task model.
>
> |         Network         | ResNet18 | ResNet34 | ResNet50 | Swin-T | Swin-S | Swin-B |
> | :---------------------: | :------: | :------: | :------: | :----: | :----: | :----: |
> | Time of task model / ms |   1.82   |   3.34   |   4.45   |  8.80  | 17.17  | 17.54  |
>
> The following shows the single-frame inference latency on an NVIDIA 4090 GPU for COCO 2017 detection and instance segmentation. Across these backbones, **the JND estimator has a nearly constant latency of approximately *19.7ms***, which remains acceptable relative to the task model.
>
> |         Network         | ResNet50 | Swin-T | Swin-S | PVTv2-b1 | PVTv2-b2 | PVTv2-b3 |
> | :---------------------: | :------: | :----: | :----: | :------: | :------: | :------: |
> | Time of task model / ms |  23.23   | 27.94  | 35.42  |  32.61   |  49.43   |  65.07   |
>
> In the future, we are committed to developing lighter and faster FeatJND estimation schemes.
>
>
> ---
> ### **Q2. The role of parameter $\alpha$ in Sec. 4.3**
>
> The parameter $\alpha$ is a scalar multiplier that controls the strength of the FeatJND-based distortion. The distorted feature is constructed as $\tilde{f} = f + \alpha \cdot \delta_{\text{JND}}$, where $\delta_{\text{JND}} = G_{\theta}(f)$ is the predicted FeatJND map. The JND map itself indicates a task-aligned distortion direction that concentrates perturbation on less task-sensitive feature components while suppressing distortion on critical ones. The role of $\alpha$ is then to scale along this direction: $\alpha = 1$ corresponds to the predicted tolerance boundary, $\alpha < 1$ stays safely within it, and $\alpha > 1$ pushes beyond it. Fig. 8 shows how task performance degrades as $\alpha$ increases, confirming that the predicted boundary provides a calibrated tolerance threshold in feature space.
>
>
> ---
> ### **Q3. Values of $\alpha$ used in Fig. 7 and Fig. 9**
>
> In both Fig. 7 and Fig. 9, we use $\alpha = 1$, meaning the FeatJND perturbation is applied at exactly the predicted tolerance boundary. This is the natural operating point since it represents the maximum distortion that the estimator considers imperceptible to the downstream task. We will state this explicitly in the figure captions in the revised version.
>
>
> ---
> ### **Q4. How the step-size map is obtained in the feature quantization setting**
>
> We describe the procedure briefly here. First, we convert the predicted FeatJND map $\delta_{\text{JND}}$ into a non-negative relative tolerance map: $s = |\delta_{\text{JND}}| / (|f| + \epsilon)$. This captures the per-token relative sensitivity. The token-wise quantization step size is then $\Delta = \lambda s$, where $\lambda$ is a global scaling factor chosen so that the expected quantization noise variance $\mathbb{E}(\Delta^{2}/12)$ equals a target $\sigma_{\text{tgt}}^{2}$. Solving this gives $\lambda = \sqrt{12 \sigma_{\text{tgt}}^{2} / \mathbb{E}(s^{2})}$. The feature is then uniformly quantized as $Q(f) = \Delta \cdot \text{round}(f / \Delta)$. The key insight is that each spatial token receives a different step size shared across all its channels, so task-sensitive tokens get finer quantization while redundant tokens tolerate coarser steps. We will make this derivation more explicit in the main text.

---

> > ### Author Rebuttal · Reviewer_Zm5r · 2026-04-02
> >
> > Thanks the authors for their responses. All my questions have been addressed. I have no further questions and maintain the positive assessment.

---

### Official Review · Reviewer_LEhk · 2026-03-12

**Soundness:** 4
**Presentation:** 4
**Significance:** 4
**Originality:** 3
**Overall Recommendation:** 5
**Confidence:** 3

**Summary:**

This paper introduces FeatJND, a method that extends the classical concept of Just Noticeable Difference (JND) from human visual perception into the domain of deep feature representations. While traditional JND modeling focuses on identifying the maximum imperceptible distortion at the pixel level for human observers, FeatJND instead operates in the latent/feature space of neural networks, seeking to characterize the maximum per-feature perturbation that can be tolerated without degrading the performance of downstream vision tasks.

The core formulation casts this as a constrained optimization problem: an estimator network learns to produce perturbation tolerance maps such that the distorted features remain functionally equivalent to clean features under task-specific discrepancy measures. For image classification, discrepancy is quantified via temperature-scaled KL divergence between softmax outputs; for detection and instance segmentation, it is measured through region-of-interest alignment derived from clean feature proposals. The magnitude of allowable distortion is assessed using Normalized Root Mean Square Error (NRMSE), and a probabilistic constraint governs the rate of acceptable task-performance violations.

The method is evaluated on ImageNet-1k for classification and COCO for detection and segmentation, using frozen pretrained backbones without any task-specific retraining. Beyond measuring feature resilience, the authors demonstrate a practical downstream application of FeatJND by using the predicted tolerance maps as step-size allocations for token quantization, where they show improvements over both random permutation and uniform global step-size baselines under equivalent noise budgets.

The principal contributions of this work are: (1) the first JND formulation explicitly targeting machine perception in the feature space rather than human visual quality; (2) a task-agnostic estimator that generalizes across classification, detection, and segmentation without architectural specificity; and (3) a demonstrated utility of FeatJND tolerance maps as a practical signal for feature-space quantization.

**Compliance With Llm Reviewing Policy:**

Affirmed.

**Final Justification:**

Following a careful review of the manuscript and the authors' comprehensive rebuttal, I strongly recommend accepting this paper. The authors provided thorough, convincing responses to all of my initial queries, which has fully resolved my concerns and reinforced my prior positive assessment of the work.

**Clarity**: The manuscript is well-structured and clearly written. The authors successfully articulate complex concepts, making the mathematical formulations and the overall architecture easy to follow.

**Soundness**: The proposed framework is technically robust. The authors effectively bridge established psychophysical principles with modern deep feature representations. The experimental design is rigorous, and the empirical results convincingly support the theoretical claims made regarding feature-level JND.

**Key Questions For Authors:**

What is the criterion for selecting the interface feature f as the network split point, and how sensitive is the method to this choice?

How does FeatJND perform under a compound noise regime, where both the input image and the feature space are simultaneously corrupted?

How should the attention map degradation observed in Figure 7 be interpreted, and does it indicate a systematic failure mode?

**Limitations:**

While the authors present a technically interesting and conceptually novel contribution, the manuscript would benefit from a more transparent discussion of its current boundaries. Specifically, the paper does not adequately address the sensitivity of the method to the selection of the interface feature f, the restriction of empirical validation to the single estimator G0, the potential degradation of tolerance maps under compound noise conditions where input images are themselves corrupted, and the spatial inconsistencies observed in the attention maps of Figure 7, which may indicate boundary conditions of the estimator that warrant further investigation.

Additionally, in alignment with the venue's expectations, a brief acknowledgment of potential societal implications, particularly the possibility that feature-space perturbation characterization could be repurposed in adversarial contexts, would strengthen the paper's overall transparency and responsibility. The authors are encouraged to address these points not as fundamental flaws, but as honest reflections of the work's current scope.

**Strengths And Weaknesses:**

Soundness
Strengths

* The optimization problem is well-formulated, with a probabilistic constraint that governs task-performance violations, providing a principled boundary between tolerable and intolerable feature perturbations

* The use of task-specific discrepancy measures is methodologically appropriate: temperature-scaled KL divergence for classification and ROI-alignment for detection/segmentation reflects genuine understanding of how each task head operates

* Evaluating on frozen pretrained backbones without retraining strengthens the claim of generalizability, as the method is not overfitted to any particular training regime

* The Gaussian noise baseline comparison with multiple seeds provides a reasonable, if minimal, control condition

Weaknesses

* The selection criterion for the interface feature f (the split point in the network) is not clearly justified; this is an important hyperparameter that could significantly affect the resulting tolerance maps, and its choice appears ad hoc

* The experimental baselines are limited; comparing only against Gaussian noise and random/global quantization strategies is insufficient to establish state-of-the-art performance. No comparison against learned or perceptually-motivated feature compression methods is provided

* The paper does not evaluate robustness under a double-noise regime (corrupted inputs combined with FeatJND perturbations), leaving an important practical scenario unaddressed

* Only the estimator G0 of FeatJND is evaluated, leaving the broader architecture of the method partially unvalidated

Presentation
Strengths

* The introduction is well-constructed, guiding the reader from the motivation of feature resilience to the limitations of existing JND and feature quality methods in a logical and accessible progression

* The loss functions and their individual components are clearly explained, with explicit reasoning provided for each design choice

* The paper situates itself clearly within two distinct bodies of related work, classical JND modeling and feature quality estimation, and articulates meaningful distinctions from both

Weaknesses

* Figure 7 is cited in Section 4.4 but appears after Figures 8 and 9 in the text flow, breaking the logical narrative and making it harder to follow the experimental analysis

* Figure 8 contains a dashed blue line that is never defined or explained in the caption or surrounding text, which is a clear presentation oversight

* The figure ordering and cross-referencing issues suggest the paper would benefit from a careful structural revision of the experimental section before publication


Significance
Strengths

*The practical application to token quantization via step-size allocation demonstrates that FeatJND tolerance maps have immediate, concrete utility beyond theoretical characterization; this downstream use case meaningfully broadens the paper's impact

* Addressing feature-space distortion tolerance is a practically relevant problem, particularly for efficient inference, model compression, and robust deployment pipelines

* The framing of "machine perception imperceptibility" as a design objective could open new directions in feature compression, adversarial robustness, and neural codec design

Weaknesses

* The scope of demonstrated applications remains narrow, quantization is the only applied use case shown, and the gains over baselines, while positive, are not accompanied by sufficient analysis of when and why FeatJND provides an advantage

* The impact on very deep or attention-based architectures is partially undermined by the attention map artifacts visible in Figure 7, where the tolerance maps shift focus away from the primary object toward background regions in some examples, raising questions about reliability at scale


Originality
Strengths

* To the best of the reviewer's knowledge, this is the first work to explicitly extend the JND framework to the machine perception domain, targeting latent feature representations rather than pixel-level human visual sensitivity, which represents a genuinely novel conceptual contribution

* The task-aligned formulation that unifies classification, detection, and segmentation under a single probabilistic tolerance framework is a creative and non-trivial synthesis

* The positioning against both HVS-based JND literature and architecture-specific feature quality models clearly delineates a gap that this work meaningfully fills

Weaknesses

* While the conceptual novelty is clear, the technical machinery (KL divergence, NRMSE, ROI alignment) draws entirely from well-established components; the originality lies primarily in their combination and motivation rather than in any novel technical derivation

* The claim of universal adaptability across architectures, as opposed to existing architecture-specific methods, is stated but not sufficiently stress-tested across diverse backbone families to be fully substantiated

---

> ### Author Rebuttal · Authors · 2026-03-30
>
> We sincerely thank the reviewer for the thoughtful feedback and the recognition of our principled formulation and practical utility. We address each weakness point below.
>
> ### **Soundness Part**
>
> ### 1. Split point selection.
>
> The split point is not ad hoc. We intentionally place it at the backbone output for classification and neck output for detection/segmentation, matching practical feature handoff settings in collaborative intelligence and edge-cloud deployment where these outputs serve as the standard transferable interface. Our goal is to study JND on this *reusable* representation, not to search for an optimal split point. We acknowledge that the tolerance maps are sensitive to this choice, as deeper layers typically tolerate larger perturbations due to higher semantic abstraction. We will explicitly discuss this sensitivity and leave systematic layer-wise analysis for future work.
>
> ---
> ### 2. Experimental baselines.
>
> We have added the image-domain JND baseline (Jin et al., 2021) adapted to feature space; please see our response to Reviewer Uimk for details and results.
>
> ---
> ### 3. Double noise regime.
>
> Thank you for your very thoughtful advice. We add experiments directly using our pretrained models as follows. We add noise with variance $\sigma$ to images with pixel values in [0,1]. An approximate value of PSNR can be estimated according to its definition. Results below show FeatJND's effectiveness with corrupted inputs.
>
> **Accurate on image classification with Swin-S**
> |NMSE|10|20|30|40|
> |-|-|-|-|-|
> |$\sigma=0.03$ ($\text{PSNR}\approx 30.46$)|||||
> |Gaussian|80.8|78.4|68.1|45.1|
> |FeatJND|81.3|81.3|81.3|81.3|
> ||||||
> |$\sigma=0.05$ ($\text{PSNR}\approx 26.02$)|||||
> |Gaussian|79.2|76.6|65.6|42.6|
> |FeatJND|79.8|79.9|79.8|79.7|
> ||||||
> |$\sigma=0.07$ ($\text{PSNR}\approx 23.10$)|||||
> |Gaussian|77.4|74.7|63.1|40.1|
> |FeatJND|78.0|78.0|78.1|77.9|
>
> **mAP@[0.5:0.95] on image detection with Swin-S+Mask-RCNN**
> |NMSE|2|3|4|5|
> |-|-|-|-|-|
> |$\sigma=0.03$ ($\text{PSNR}\approx 30.46$)|||||
> |Gaussian|44.2|42.6|40.6|36.5|
> |FeatJND|45.9|45.6|45.0|44.1|
> ||||||
> |$\sigma=0.05$ ($\text{PSNR}\approx 26.02$)|||||
> |Gaussian|44.2|40.6|38.5|34.8|
> |FeatJND|43.8|43.4|42.7|42.1|
> ||||||
> |$\sigma=0.07$ ($\text{PSNR}\approx 23.10$)|||||
> |Gaussian|40.0|38.6|36.6|33.0|
> |FeatJND|41.4|41.1|40.5|39.8|
>
> ---
> ### 4. Estimator Variants.
>
> Our goal here is to validate the FeatJND formulation, rather than to perform a broad estimator architecture study. We therefore use $G_\theta$ as a simple and lightweight instantiation, so that the gains are less likely to be explained by estimator capacity alone. We agree that evaluating more estimator variants would further strengthen the paper, and we have also provided architecture and hyperparameter ablations in our response to Reviewer Uimk.
>
>
>
> ### **Presentation Part**
>
> ### 5--7. Figure ordering, dashed line in Fig. 8, cross-referencing.
>
> We will fix the figure ordering (Figs. 7–9), clarify that the dashed lines in Fig. 8 are visual aids indicating representative performance-drop levels (not used in training or evaluation), and revise the cross-references.
>
>
> ---
> ### **Significance Part**
>
> ### 8.  Application scope and analysis.
>
> We will clarify that our current experiments center on quantization as a proxy application. FeatJND helps because it learns a task-aligned, heterogeneous token-wise tolerance. Under matched noise budgets, it allocates larger steps to less task-sensitive components, whereas random and uniform baselines break or ignore this allocation.
>
>
> ---
> ### 9. Artifacts in Fig. 7.
>
> We clarify that Fig. 7 visualizes **attribution maps** before and after applying FeatJND-based distortion, not the predicted tolerance maps themselves. Our method optimizes task-output consistency, not attribution-map preservation, so some local attribution redistribution can occur without implying a systematic failure mode. Quantitative evidence (Figs. 4–6, 8) confirms performance is preserved. We will clarify this.
>
> ---
> ### **Originality Part**
>
> ### 10--11. Novelty and cross-architecture adaptability.
>
> We agree that FeatJND's novelty lies in the new problem formulation and method as a whole, operationalizing JND in deep feature space with task-aligned objectives, rather than in any individual component. Regarding adaptability, we acknowledge that "universal" may overstate the current scope. Our intended claim is that FeatJND is not tied to a specific backbone design: it is validated across ResNet, Swin, and PVTv2 families on three tasks without retraining downstream models. We will revise the wording accordingly, and broader backbone coverage is a natural future direction.
>
> ---
> ### **Limitations and Societal Implications**
> We will expand the Limitations section to discuss split-point sensitivity, the single $G_\theta$ estimator, and compound noise behavior. We will also explicitly acknowledge the societal risk of FeatJND being repurposed for adversarial attacks.

---

> > ### Author Rebuttal · Reviewer_LEhk · 2026-04-03
> >
> > I thank the authors for their detailed rebuttal. All of my concerns have been satisfactorily addressed. I have no further questions and maintain my positive assessment of the manuscript.

---

### Official Review · Reviewer_Uimk · 2026-03-13

**Soundness:** 3
**Presentation:** 3
**Significance:** 3
**Originality:** 2
**Overall Recommendation:** 4
**Confidence:** 3

**Summary:**

This paper proposes FeatJND, which extends the just noticeable difference (JND) concept from the human visual system to deep visual feature space. The core idea is to learn a token-level tolerance map $\delta_{JND} = G_\theta(f)$ that allows maximum perturbation in feature space while keeping the outputs of downstream tasks nearly unchanged. This is formulated via a chance-constrained optimization and a Lagrangian relaxation into a two-term loss, implemented with a small CNN estimator $G_\theta(\cdot)$. The method demonstrates superiority over Gaussian noise across diverse backbones (ResNet, Swin, and PVTv2) on the ImageNet classification, COCO object detection, and instance segmentation. This paper also presents token-wise feature quantization as a practical proxy application.

**Compliance With Llm Reviewing Policy:**

Affirmed.

**Final Justification:**

The authors have adequately addressed my questions. Accordingly, I adjusted my score.

**Key Questions For Authors:**

1. Can you provide ablation results for $G_\theta$'s architecture ($N_{res}$, kernel size) and key hyperparameters ($T$, $\lambda_t$, $\tau$)?
2. Can you compare against a gradient-magnitude sensitivity map as a training-free task-aware baseline, to isolate whether gains stem from the learned formulation specifically?
3. Can you apply image-domain JND methods (Jin et al., 2021; Zhang et al., 2021a) directly to feature space as a baseline, to demonstrate that the feature-domain extension is non-trivial?

**Limitations:**

The limitations, e.g., task/backbone-specific retraining overhead, should be discussed more explicitly.

**Strengths And Weaknesses:**

**Strengths**
- The problem is well-motivated. Feature-space JND is an important concept in real systems such as collaborative intelligence, feature compression, and edge-cloud deployment, yet has not been systematically studied. This paper has value as a first attempt.

- The token-wise quantization experiment is a concrete demonstration of practical utility, showing consistent gains over random permutation and global uniform baselines across classification, detection, and segmentation.

**Weaknesses**

- Comparison for evaluation is insufficient, for two distinct reasons. (1) The paper cites image-domain JND methods with the same objective (Jin et al., 2021; Zhang et al., 2021a). However, applying them to feature space appears to require only a domain change rather than a new technical contribution. Without comparing to these methods, it is unclear whether the feature-domain extension is truly non-trivial. (2) A gradient-magnitude sensitivity map ($|\partial \mathcal{L}/\partial f|$) is a training-free alternative that the authors could construct themselves. Without it, it is impossible to determine whether FeatJND's gains stem from the learned formulation or simply from any task-aware perturbation allocation.

- The quantization baselines (random permutation, global uniform) are too weak: the random baseline is effectively an ablation of FeatJND's allocation strategy rather than an independent competing method.

- Ablation studies are entirely absent. Sensitivity to $G_\theta$'s architecture ($N_{res}$, kernel size), $T$, $\lambda_t$, and $\tau$ is never analyzed, limiting reproducibility.

---

> ### Author Rebuttal · Authors · 2026-03-30
>
> We thank the reviewer for the positive assessment. We organize responses by the reviewer's questions and will update our revision accordingly.
>
> ### Responses to Questions
>
> ### **Q1. Ablation studies.**
>
> $\tau$ appears only in the intermediate derivation (Eq. 3--4) and is absorbed into the fixed $\lambda_t$ in the final training objective (Eq. 8), so it does not appear as a separate tunable quantity in our implementation.
>
> We provide ablation results below. Since we cannot attach images, we use some key data points instead.
>
> **Ablations on Swin-S for image classification**
> |$\lambda_\text{t}$ / NRMSE|10|20|30|40|
> |-|-|-|-|-|
> |10|82.0|76.0|65.7|53.7|
> |30|82.8|82.7|82.6|82.4|
> |**50**|82.8|82.7|82.6|82.5|
> |70|82.8|82.7|82.6|82.5|
> ||||||
> |$T$|||||
> |1|82.6|82.0|80.8|78.7|
> |**4**|82.8|82.7|82.6|82.5|
> |8|82.8|82.8|82.6|82.4|
> ||||||
> |$N_{res}$|||||
> |1|82.8|82.7|82.6|82.5|
> |2|82.8|82.7|82.6|82.5|
> |**3**|82.8|82.7|82.6|82.5|
> |4|82.7|82.7|82.6|82.4|
> ||||||
> |$K_{size}$|||||
> |**1**|82.8|82.7|82.6|82.5|
> |3|82.8|82.7|82.5|82.2|
> |5|82.7|82.5|82.2|81.8|
>
>
> **Ablations on Swin-S+Mask-RCNN for object detection (mAP[0.5:0.95])**
> |$\lambda_\text{t}$ / NRMSE|2|3|4|5|
> |-|-|-|-|-|
> |10|46.9|44.7|41.8|38.3|
> |100|48.2|47.9|47.6|47.0|
> |**200**|48.4|48.2|47.9|47.5|
> |300|48.4|48.3|48.0|47.7|
> ||||||
> |$T$|||||
> |1|47.7|46.9|46.0|45.1|
> |**4**|48.4|48.2|47.9|47.5|
> |8|48.4|48.1|47.9|47.7|
> ||||||
>
> Results for $N_{res}$ and $K_{size}$ are similar to the classification ablations and omitted for brevity.
>
> As shown above, unless $\lambda_t$ is extremely small, all factors have a very small impact on task performance.
>
>
> ---
> ### **Q2. Gradient-magnitude sensitivity map.**
>
> We agree that a gradient-magnitude map is a meaningful task-aware reference. We conducted direct comparisons on ResNet-34 and Swin-S, reporting Top-1 accuracy at matched NRMSE levels:
>
> **Comparison on ResNet34 for image classification**
> |NRMSE|5|10|15|20|
> |-|-|-|-|-|
> |Grad Mag|72.3|69.4|64.7|57.1|
> |**FeatJND**|73.3|73.2|73.1|73.1|
>
> **Comparison on Swin-S for image classification**
> |NRMSE|5|10|15|20|
> |-|-|-|-|-|
> |Grad Mag|82.1|79.2|71.9|58.1|
> |**FeatJND**|82.8|82.8|82.8|82.7|
>
> FeatJND outperforms the grad-mag baseline. Notably, the grad-mag map **requires ground-truth** labels and a **per-sample backward** pass through the task head to compute $|\partial \mathcal{L}/\partial \mathbf{f}|$ at every inference. FeatJND predicts the tolerance map with a single forward pass using only the feature tensor, without labels, model weights, or gradients at inference time.
>
> **Comparison of requirements between the grad-mag map and FeatJND**
> |Method|Labels/annotations needed|Downstream model weights needed|Downstream gradients needed|
> |-|-|-|-|
> |Grad Mag|Yes*|Yes|Yes|
> |**FeatJND**|No|Training: Yes; Inference: No|Training: Yes; Inference:No|
>
> \* For the standard supervised construction based on the downstream task loss, i.e., $|\partial \mathcal{L}/\partial \mathbf{f}|$.
>
>
> ---
> ### **Q3 More comparisons.**
>
> ### **Q3-1. Direct comparison with Jin et al., 2021**
>
> We additionally conducted experiments by directly using the method of Jin et al., 2021 as a baseline on both Swin-S and ResNet-34. Since the method of Jin et al. is specifically designed for the image classification task, we conduct this comparison on classification only. For a fair comparison, we adapted their original multi-network design to the same single-network setting as ours, while keeping their predictor architecture and loss functions. The corresponding results are shown below.
>
> **Comparison on ResNet34**
> |NRMSE|3|6|9|12|
> |-|-|-|-|-|
> |Jin et al.|57.4|45.4|39.0|35.4|
> |**FeatJND**|73.2|73.1|73.0|72.9|
>
> **Comparison on Swin-S**
> |NRMSE|10|20|30|40|
> |-|-|-|-|-|
> |Jin et al.|80.5|76.3|72.4|69.0|
> |**FeatJND**|82.8|82.7|82.6|82.5|
>
> As can be seen, directly transferring the image-domain JND design to feature space does not match the performance of FeatJND, suggesting that the feature-domain extension is not a trivial domain substitution.
>
> ### **Q3-2. Comparison with Zhang et al., 2021a**
>
> We respectfully note that Zhang et al., 2021a studies JRD (Just Recognizable Distortion) for machine-oriented coding, where JRD is a discrete QP label predicted by an ensemble-learning method to control compression. In contrast, FeatJND models a continuous per-feature tolerance map in deep feature space. Thus, their work is better regarded as related work rather than a baseline for our setting.
>
>
> ---
> ### **Quantization baseline**
> The random baseline proves spatial allocation's value. We also evaluated an energy-based baseline, which fails in detection: at $\sigma=4$, on Swin-S+Mask-RCNN, its mAP plummets by over 45 points, whereas FeatJND degrades by less than 2 points.
>
> ### **Model retraining overhead**
> We agree that training $G_\theta$ per task/backbone introduces overhead. We will discuss this limitation in the revision. As noted in Sec. 5, our future work aims to develop a task- and model-general predictor to mitigate this.

---

> > ### Author Rebuttal · Reviewer_Uimk · 2026-04-01
> >
> > The authors have adequately addressed my questions. Accordingly, I will adjust my score.

---

### Decision · Program_Chairs · 2026-04-30

**Decision:**

Accept (spotlight)

**Comment:**

This paper introduces FeatJND, a task-aligned formulation of just noticeable difference in deep visual feature space, together with an estimator for per-feature perturbation tolerance and an application to token-wise quantization. The reviewers agreed that the paper studies a novel and well-motivated problem. They found the formulation principled, the presentation clear, and the empirical study broad across classification, detection, and segmentation. Several reviewers also highlighted the practical value of the quantization experiments and viewed the extension of JND from pixel space to feature space as the paper’s main conceptual contribution.

In the first round, the main questions concerned the strength of the baselines and ablations, the choice of the interface feature and sensitivity to architectural or hyperparameter choices, and several implementation details such as inference overhead, estimator inputs, and the construction of the quantization step-size map. After carefully reading the rebuttal and discussion, I am satisfied that these concerns were addressed adequately. The authors clarified the methodological choices, provided the missing details, and responded convincingly to questions about comparison, sensitivity, and practical applicability. Reviewers explicitly noted that their concerns had been resolved and maintained positive recommendations, with one reviewer raising the score after rebuttal.

I therefore recommend strong acceptance. The paper presents a clear and original perspective on feature-space quality modeling and supports it with solid empirical evidence.